# An NMDA receptor-dependent mechanism for subcellular segregation of sensory inputs in the tadpole optic tectum

Ali S Hamodi[†], Zhenyu Liu, Kara G Pratt*

Department of Zoology and Physiology and Program in Neuroscience, University of Wyoming, Laramie, United States

**Abstract** In the vertebrate CNS, afferent sensory inputs are targeted to specific depths or layers of their target neuropil. This patterning exists ab initio, from the very beginning, and therefore has been considered an activity-independent process. However, here we report that, during circuit development, the subcellular segregation of the visual and mechanosensory inputs to specific regions of tectal neuron dendrites in the tadpole optic tectum requires NMDA receptor activity. Blocking NMDARs during the formation of these sensory circuits, or removing the visual set of inputs, leads to less defined segregation, and suggests a correlation-based mechanism in which correlated inputs wire to common regions of dendrites. This can account for how two sets of inputs form synapses onto different regions of the same dendrite. Blocking NMDA receptors during later stages of circuit development did not disrupt segregation, indicating a critical period for activity-dependent shaping of patterns of innervation.

*For correspondence: Kpratt4@uwyo.edu

Present address: [†]Department of Neurobiology, Yale University School of Medicine, New Haven, United States

Competing interests: The authors declare that no competing interests exist.

## Introduction

In most regions of the vertebrate CNS, afferent axonal inputs form topographic maps within specific layers of their target region. This laminar patterning of maps allows for multiple sets of afferent inputs to coexist across the same two dimensional plane of a common target, conducive to integration. This patterning motif is found in many brain structures that receive multiple sets of afferent input. For example, the different sensory inputs that are received and processed by the optic tecta of birds (*Acheson et al., 1980*; *Yamagata et al., 1995*), fish (*Xiao et al., 2005*; *Xiao and Baier, 2007*), and amphibians (*Harris, 1982*, *1983*; *Udin and Fawcett, 1988*; *Deeg et al., 2009*; *Hiramoto and Cline, 2009*), and the superior colliculus of mammals (*May, 2006*; *Wallace et al., 1993*; *Cang and Feldheim, 2013*; *Inayat et al., 2015*), form distinct layers across the laminar axis of their target neuropil. In the lateral geniculate nucleus (LGN), retinal ganglion cell input is segregated into eye-specific layers (*Rakic, 1976*; *Linden et al., 1981*; *Shatz, 1983*), and in the hippocampus, entorhinal cortical input forms synaptic connections onto specifically the distal regions of hippocampal pyramidal neurons, while the commissural inputs from within the hippocampus synapse onto the proximal portions of the same dendrites (*Supèr and Soriano, 1994*). Lamination is also a means for precise targeting of axons to their specific subcellular synaptic targets because, by being restricted to a particular layer, presynaptic inputs form synapses with only the section of dendrite that is within that layer (*Huberman et al., 2010*; *Yamagata and Sanes, 1995*; *Sanes and Yamagata, 1999*). With the exception of the eye-dependent segregation of inputs to the LGN, which are initially overlapping with one another and then become segregated or refined into layers by activity-dependent mechanisms, most lamination of inputs appears to exist ab initio, from the beginning, before sensory-driven activity is present, and therefore is considered to be an activity-independent process (*Huberman et al., 2010*; *Xiao and Baier, 2007*; *Xiao et al., 2011*; *Takahashi et al., 1999*).

Nevertheless, the subcellular precision in which synapses form onto their postsynaptic targets suggests that, similar to the refinement of the topographic map, activity-dependent mechanisms may also be important, once activity is present.

Here we test if the subcellular targeting of synapses is achieved via a similar activity-dependent mechanism that underlies refinement of the topographic map. For this we use the *Xenopus* tadpole optic tectum as a model system, a relatively simple and well-studied multisensory integration center. The optic tectum receives direct visual input from the retinal ganglion cells in the eye and mechanosensory input that enters the tectum via the brainstem. Both sets of afferent sensory inputs form topographic maps parallel with the surface of the neuropil, (*Sperry, 1963*; *Holt and Harris, 1983*; *Sakaguchi and Murphey, 1985*; *Udin and Fawcett, 1988*; *Hiramoto and Cline, 2009*). These maps are segregated from one another such that visual inputs innervate the distal-most neuropil and mechanosensory inputs innervate the proximal-most layer, the region closest to the somata of the tectal neurons (*Hiramoto and Cline, 2009*; *Hamodi and Pratt, 2015*). Both sets of sensory inputs are glutamatergic and activate postsynaptic responses via AMPA and NMDA receptors expressed on tectal neuron dendrites (*Wu et al., 1996*; *Deeg et al., 2009*). At the stages studied here, deep-layer tectal neuron dendrites extend the entire length of the laminar axis (*Lázár, 1973*; *Wu and Cline, 2003*) and it is well established that single tectal neurons are directly innervated by both visual and non-visual inputs (*Pratt and Aizenman, 2009*; *Deeg et al., 2009*; *Hiramoto and Cline, 2009*).

We take advantage of a modified whole brain preparation that allows for the spatial pattern of functional visual and non-visual mechanosensory inputs to be measured electrophysiologically, resulting in a high resolution map of synaptic activity generated at different points across the laminar axis of the neuropil (*Hamodi and Pratt, 2015*). Our results show that the normal segregation between the visual and mechanosensory inputs across the tectal neuron dendrites relies on NMDA receptor-dependent activity, and suggests that the subcellular targeting of axons to a particular region of dendrite is achieved via a correlation-based mechanism that leads to the congregation of the afferent inputs with correlated firing patterns and the elimination of local non-correlated inputs.

## Results

### Visual and non-visual afferent inputs innervate distinct lamina of the optic tectum

It has been previously determined that, in the tadpole optic tectum, RGC afferent axons are targeted to the distal (lateral) region of the laminar axis while the non-visual mechanosensory inputs that enter the tectum via the hindbrain (HB), and so called 'HB inputs', target the proximal (medial) region (*Hiramoto and Cline, 2009*; *Deeg et al., 2009*); *Figure 1A*). First, we confirmed this finding by labeling the RGC axons of stage 48/49 tadpoles with DiD (green), HB inputs with DiI (red), and imaging their terminations in the tectum (*Figure 1B–D*). We observed that, as previously reported, the two different sensory inputs innervate distinct, non-overlapping lamina in the neuropil with RGC inputs confined to distal lamina, HB inputs to the proximal. On the postsynaptic side, deep-layer tectal neurons display monosynaptic responses to both RGC and HB input activation (*Hamodi and Pratt, 2015*; *Deeg et al., 2009*), indicating that the RGC and HB inputs are targeted to different regions of the same tectal neuron dendrite, and that dendrites span essentially the full length of the laminar axis of the neuropil. Both evoked responses include a temporally distinct monosynaptic response due to the direct synaptic activation by the afferent input, followed by the polysynaptic portion of the response due to the activation of local microcircuitry within the tectum. An example of a whole cell recording of typical RGC- and HB-evoked responses from a single deep-layer tectal neuron of a stage 49 tadpole is shown in *Figure 1E*.

Although imaging these axons reveals a distinct stratified and non-overlapping pattern formed by the different axon terminals across the laminar axis, this method does not tell us about the *functional* pattern of synaptic input. To address this, we used a modified whole brain preparation that allows direct access to the entire distal-proximal axis of the neuropil while, like the traditional whole brain preparation, retaining both the RGC and HB inputs. It was previously established that the pattern and strength of RGC and HB-evoked responses recorded from tectal neurons of the modified preparation do not differ from that recorded in the traditional preparation, indicating that RGC and HB inputs are not severed in the modified brain preparation (*Hamodi and Pratt, 2015*). Hence, the

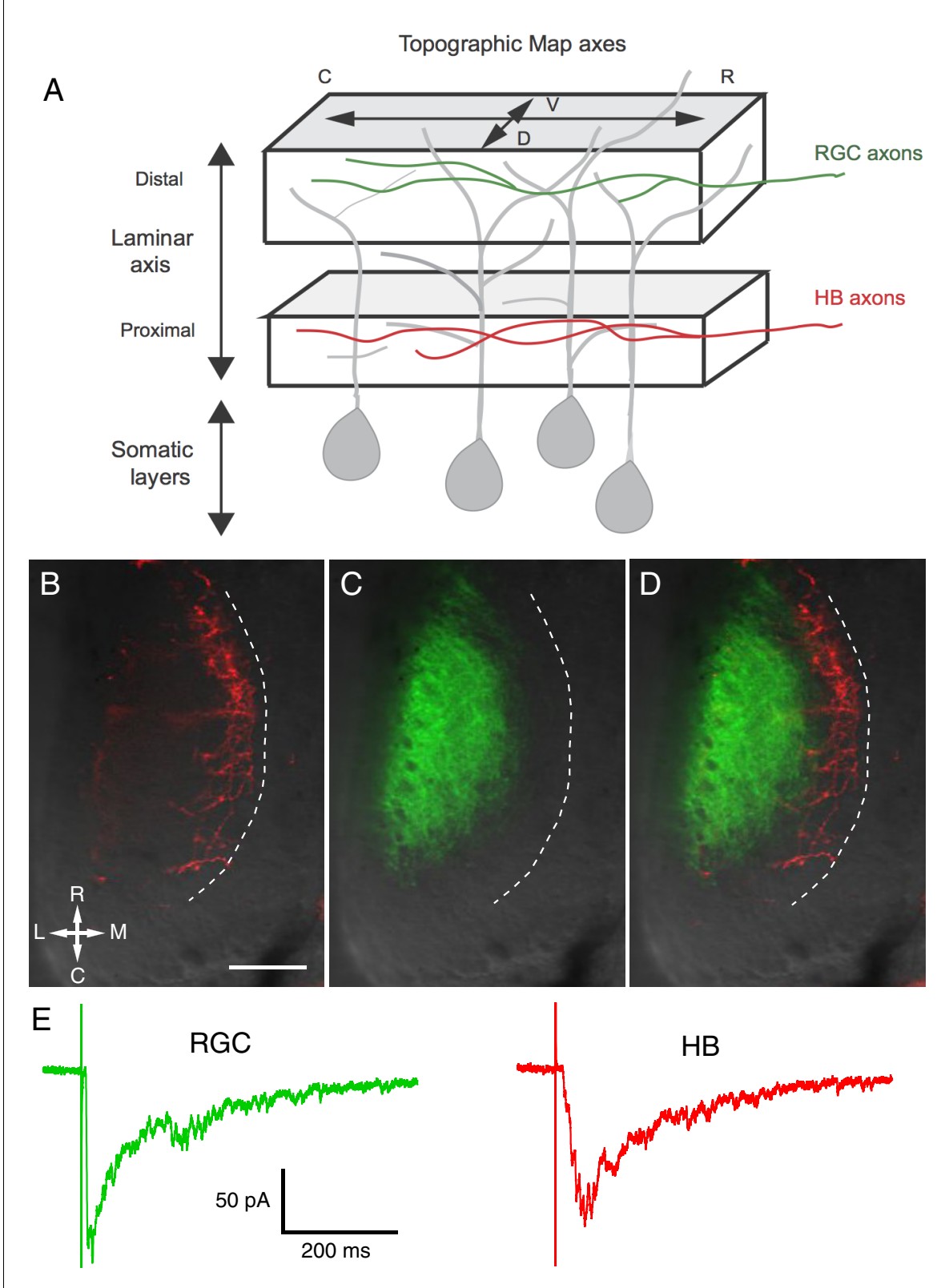

**Figure 1.** Visual and non-visual afferent inputs innervate different layers of the optic tectum neuropil and form synapses onto specific regions of tectal neuron dendrites. (**A**) Schematic of the optic tectum neuropil showing RGC inputs innervating the more distal or region of the neuropil and mechanosensory (HB) inputs innervating the more proximal region. The dendrites of the tectal neurons span the entire laminar axis such that an individual dendrite receives synaptic input from both sensory inputs. Both sets of afferent inputs form topographic maps across the rostro-caudal (R-C)

*Figure 1 continued on next page*

*Figure 1 continued*

and dorsal-ventral (D-V) axes. (**B**) A brightfield image of a single stage 49 optic tectum overlaid with fluorescent images to show DiI-labeled ipsilateral and contralateral hindbrain input (red), (**C**) DiD-labeled contralateral RGC input (green), and (**D**) merged image of B and C. The laminar axis lies along the lateral (L) to medial (M) axis. Notice that the RGC and HB inputs are segregated across this axis. The dashed line indicates the border between the somatic layer and the neuropil. (**E**) Whole cell recordings from a single tectal neuron in response to activation of (left) RGC inputs and (right) mechanosensory (HB) inputs. Note that both inputs evoke a monosynaptic response followed by polysynaptic activity, indicating that single neurons receive direct monosynaptic input from both sensory modalities and that both inputs activate local polysynaptic activity within the tectum. Scale bar is 50 μm. R: rostral, C: caudal, L: lateral, M: medial.

modified brain preparation allows for the activation of both the RGC and HB inputs while recording resulting synaptic field potentials (FPs) at different points along the distal-proximal laminar axis (*Figure 2A,B*). Peak FP amplitudes, which reflect the absolute strength of the postsynaptic response at that particular position in space, were measured and then normalized by setting the maximal peak FP amplitude to 1.0 for each tectum, allowing for the pattern of FP responses to be averaged across multiple tecta. In addition, current source densities (CSDs) were calculated from the spatial pattern of FPs. The CSDs, derived by calculating the second spatial derivative, subtracts the local background depolarization thus revealing the precise location of the sensory evoked response. Calculating CSDs from field potential recordings has been used to analyze the spatial organization of inputs in the adult frog optic tectum (*Chung et al., 1974*; *Nakagawa et al., 1997*; *Nakagawa and Matsumoto, 1998*, *2000*). The CSDs are then converted to image plots. *Figure 2* shows an example of RGC-evoked FPs recorded from different points along the laminar axis of the tectum of a stage 49 tadpole (*Figure 2B*). From the FPs, CSDs are calculated (*Figure 2C*), and the corresponding image plot is shown in *Figure 2D*. The major sink appearing at the far distal end of the neuropil is a characteristic of the RGC-evoked image plot observed for normal tecta. The HB-evoked response is characterized by the maximum FP and CSD sink occurring at the proximal end of the laminar axis (see *Figure 2—figure supplement 1* for an example of an HB-evoked FPs, CSDs, and resulting image plot). Throughout this study, the location of both the maximum FP amplitude and CSD sink are reported as the distance from distal edge. In addition to revealing the spatial pattern of synapses formed by the afferent sensory inputs, our data include the sensory-evoked spatial pattern of synaptic connections of the local microcircuity that generates the polysynaptic portion of the response (*Figure 2D*). It is interesting to note that the microcircuitry also displays a spatially organized, stratified pattern that closely matches that of the monosynaptic pattern.

Having established that RGC and HB inputs are indeed segregated across the laminar axis of the tadpole optic tectum and a method to examine at high resolution the spatial pattern of functional synaptic inputs made by the two different sensory inputs, we next determine if the pattern of innervation requires the presence of both of these inputs.

## Monocular enucleation results in invasion of HB axons into dendritic territory normally occupied by RGC axons

It has previously been shown that the RGC and HB afferent inputs begin innervating the tectum simultaneously (*Hiramoto and Cline, 2009*), suggesting a reciprocal interaction between them. If such an interaction exists, it would be expected that removing one of the two sets of inputs would affect the remaining input. To test this, we eliminated the RGC input to one of the two tectal lobes by surgically removing the contralateral eye at stage 34, the stage when RGC axons have just begun to exit the eyecup (*Hocking and Mcfarlane, 2007*; *de La Torre et al., 1997*). During the early larval stages of tadpole development which we focus on in this study, the major source of visual input received by the tectum is from the contralateral eye. Thus removing the contralateral eye is an effective method to eliminate the majority of RGC input received by that tectum. Occasionally, sparse ipsilateral RGC input has been observed in tadpoles at these stages (*Munz et al., 2014*); and we, too, have observed the presence of sparse ipsilateral input innervating control tecta and tecta whose contralateral input has been removed via enucleation (data not shown). Hence, removing the contralateral input removes either all or a vast majority of visual input received by the contralateral optic tectum. Once tadpoles reached stage 49, the stage when the two sensory circuits are normally well-established and functional, we recorded HB-evoked FPs from (1) the tectum that is devoid of

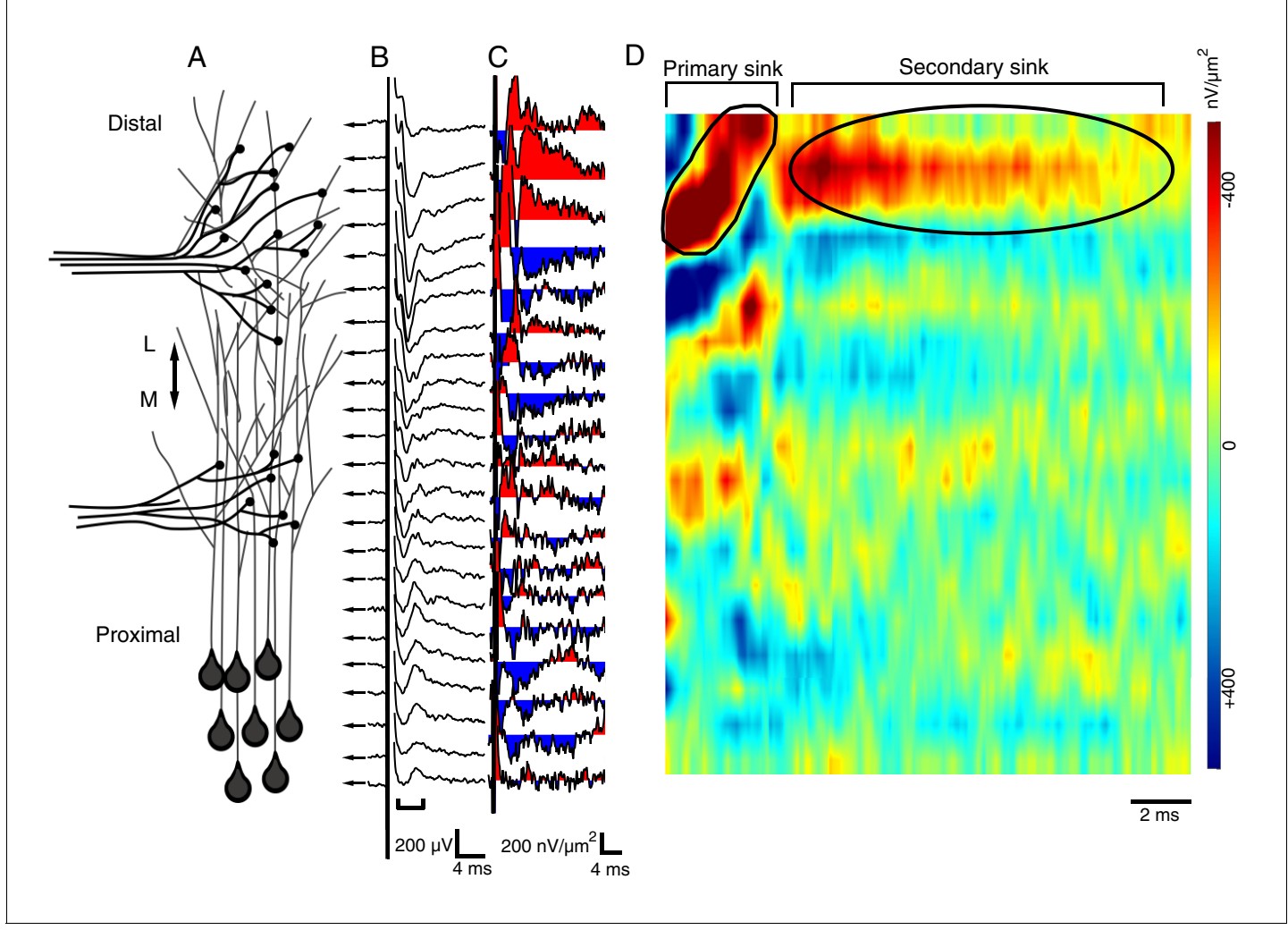

**Figure 2.** The spatial pattern of synaptic input is determined by recording RGC- and HB-evoked synaptic field potentials at equidistant points along the distal-proximal neuropil. (A) Simplified schematic of the tectal neuropil showing RGC (distal) and HB (proximal) axonal inputs onto tectal neuron dendrites. RGC and HB-evoked synaptic field potentials are recorded every 10 μm along the distal-proximal axis of the neuropil. (B) An example of RGC-evoked FPs recorded from across the distal-proximal axis of a single optic tectum. (C) Corresponding current-source density (CSD) profile derived using spatial differentiation grid of 20 μm. (D) Corresponding image plot generated using same CSD data. Red indicates current sink, blue indicates current source. Main RGC-evoked sink is localized at the distal end of the neuropil (circled primary sink). A small, more proximal sink is also commonly observed. Note that the main primary sink is followed by the recurrent portion of the response (circled secondary sink), which consistently appears in the same region as the primary sink. L: lateral, M: medial.

The following figure supplement is available for figure 2:

**Figure supplement 1.** Spatial pattern of synaptic input is determined by recording RGC- and HB-evoked synaptic field potentials at equidistant points along the distal-proximal neuropil, an example of an HB-evoked response.

contralateral RGC input (DCR) and, (2) as an internal control, the tectum (of the same brain) that receives normal, unaltered compliment of RGC input (*Figure 3A*), For each recorded tectum, the peak HB-evoked FP amplitudes were normalized and plotted as a function of the distance from the distal-most point of the axis. An average spatial FP profile was generated for control and DCR tecta by averaging the individual plots for control and DCR tecta. Compared to control, we found that the averaged spatial pattern of HB-evoked FPs in the tecta devoid of contralateral RGC input was shifted distally (*Figure 3B*; distance from distal edge of average HB-evoked FP maximum peak amplitude for control tecta: 110 μm, n = 10, for DCR tecta: 80 μm, n = 12). In addition, we also

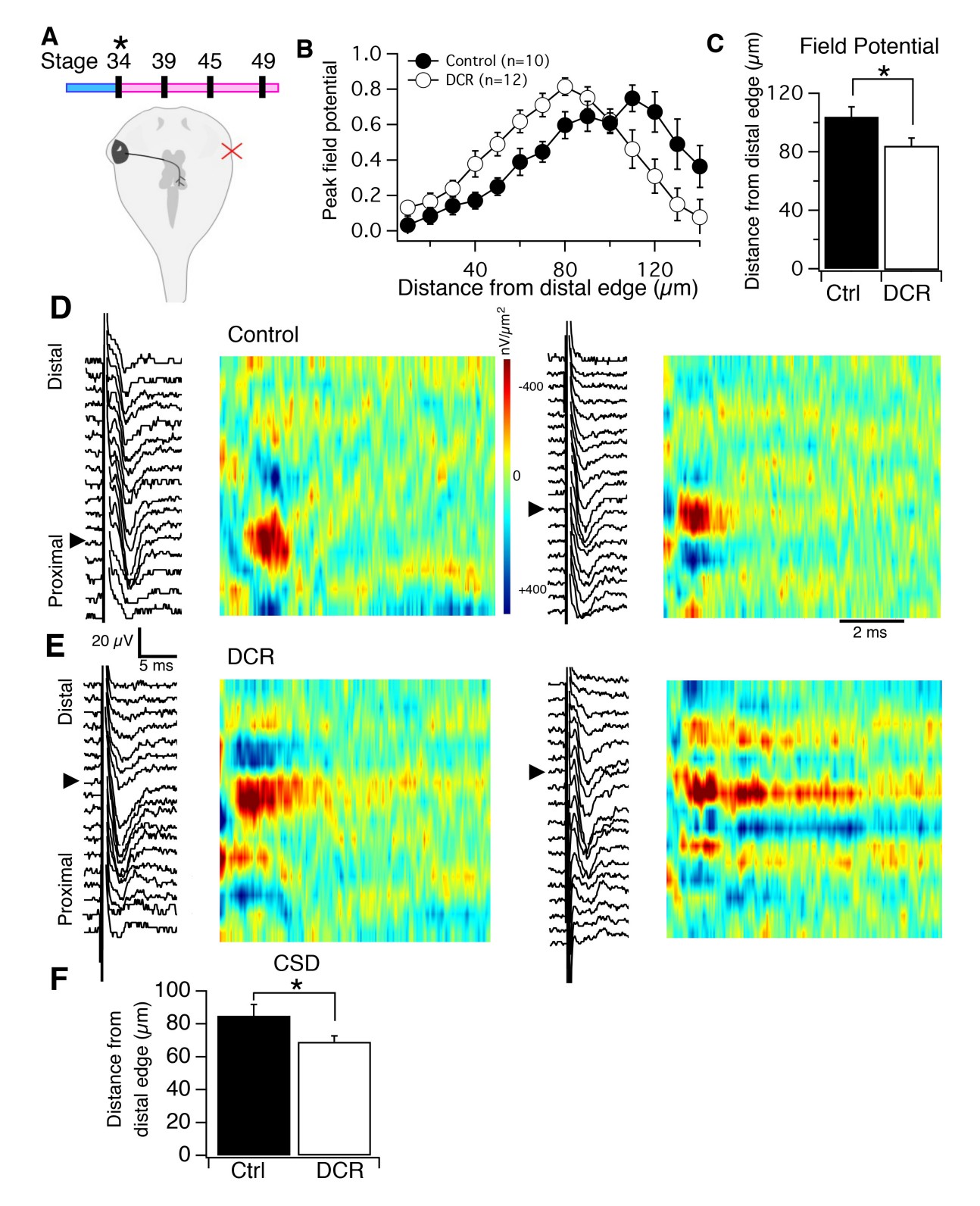

**Figure 3.** Monocular enucleation at stage 34 shifts HB inputs towards the distal neuropil. (**A**), *top*) Timeline of when enucleation and then recording was carried out, and schematic showing monocular enucleation resulting in one tectal lobe (left) devoid of contralateral RGC axons (DCR) and the other tectal lobe (right) receiving normal contralateral RGC input. (**B**) Average normalized peak FPs of HB-evoked sinks from control tectal lobe (black) and the DCR tectal lobe (red) from the same brains, n = number of tecta. (**C**) Average distance from the distal edge at which the largest HB-evoked FP

*Figure 3 continued on next page*

*Figure 3 continued*

occurred in the control (black) and DCR (white). (**D**) HB-evoked FPs from two control (i.e. tectum with normal RGC inputs) with corresponding image plots. Note that the main HB-evoked sink is localized to the proximal region of the laminar axis in both examples. (**E**) HB-evoked FPs from two DCR tecta with corresponding image plots. Note that in both image plots the main HB-evoked sink has shifted to a more distal region along the laminar axis. Also notice that the pattern appears more dispersed along the axis in the example on the right. (**F**) Average distance from distal edge at which the largest HB-evoked CSD sink occurred in the control (black) and DCR (red). For all figures: * = p<0.05, ** = p<0.01, and *** = p<0.001.

identified, for each tectum, the position across the axis where the absolute maximum HB-evoked FP amplitude occurred and found that it occurred at a significantly more distal location in the DCR tecta relative to control tecta (*Figure 3C*; average distance from distal edge where HB-evoked FP amplitude peaks for control tectum: 104 ± 6.7 µm, n = 10 tecta, 95% Confidence Interval (CI): upper limit = 117.13 µm, lower limit = 90.87 µm; DCR tectum: 84.17 ± 5.29 µm, n = 12 tecta; 95% CI: upper limit = 94.53 µm, lower limit = 73.6 µm p=0.028, unpaired t-test). This difference in the pattern of HB afferent innervation between the control and DCR tecta is reflected in the corresponding HB-evoked CSD image plots (*Figure 3D,E*). For controls, the major HB-evoked current sinks consistently appear in the proximal end of the laminar axis (*Figure 3D*) as predicted by the axon imaging experiments (*Figure 1B–D*). In the DCR tecta, however, the major HB-evoked current sinks appear at more distal locations along the laminar axis, and in some cases appeared to be widely spread out across the entire length of the laminar axis (*Figure 3E*). Averaging the distances from the distal edge at which the major HB-evoked CSD sink occurred for both control and DCR tecta indicated, similar to the FP data, that absence of RGC input causes HB inputs to innervate more distal regions of the laminar axis compared to control (*Figure 3F*; for control tectum, the major HB-evoked CSD sink occurred, on average, at 85 ± 6.9 µm from the distal edge, n = 10 tecta, 95% Confidence Interval (CI) upper limit = 98.5 µm, lower limit = 71.48 µm; for DCR tectum: 69.2 ± 3.6 µm, n = 12 tecta; 95% CI: upper limit = 76.26 µm, lower limit = 62.14 µm p=0.04, unpaired t-test.) Thus, the absence of contralateral RGC input appears to cause the pattern of HB innervation to shift more distally, and to permit a portion of HB inputs to trespass into to distal territory and form functional synaptic connections with more distal regions of tectal dendrites.

To determine if the observed shift in the pattern of synaptic input formed by the mechanosensory HB inputs is also reflected by changes in the position of their axon terminals, we injected DiD (green) into the intact eye and DiI (red) into both sides of the HB. As predicted by the CSD and FP data, we observed that HB axonal inputs in the DCR tecta appeared less focused in space and extended into the vacant dendritic region that is normally occupied by RGC axons, whereas the control tectum showed a typical confined band of HB axons along the proximal part of the laminar axis (*Figure 4A–C*). *Figure 4D* shows the fluorescent intensity profiles for the HB axons innervating the DCR and control tectal lobes shown in panel 4A. For the DCR lobe, the intensity of fluorescence, which corresponds to axon density, peaks at a more distal location compared to control. Peak fluorescence of control HB axons was found to occur, on average, at 88.6 ± 2.9 µm from the distal edge of the tectum (*Figure 4E*; n = 4 tecta, 95% CI upper limit = 97.9 µm, CI lower limit = 79.2 µm), while the peak fluorescence observed for HB axons innervating a DCR tectum occurred at 65.8 ± 4.3 µm from the distal edge (n = 4 tecta, 95% CI upper limit = 79.4 µm, CI lower limit = 52.1 µm). The difference in locations of peak HB axon fluorescence between control and DCR tecta was statistically significant (p=0.004, unpaired t-test). Combined, these data establish that the absence of the contralateral RGC input does not appear to completely unleash the HB inputs, as these inputs are still targeted to, and form synapses within, their normal target lamina. We also observed that the absence of RGC input does not result in HB inputs taking over the entire distal lamina, but rather it induces them to take over a fraction of the synaptic sites beyond their distal border, sites that would normally be occupied by retinotectal synapses. This suggests a competition-based interaction at the border/interface between the distal and the proximal portion of the laminar axis. Normally, when both afferent inputs are present, this competition could focus the different sensory inputs on their respective lamina, similar to the way activity-driven competition between RGC axons refines the topographic map. Because refinement of the topographic map is known to be NMDA receptor (NMDAR)-dependent, we tested the role of NMDARs in the segregation of the RGC and HB afferent inputs to their respective laminae.

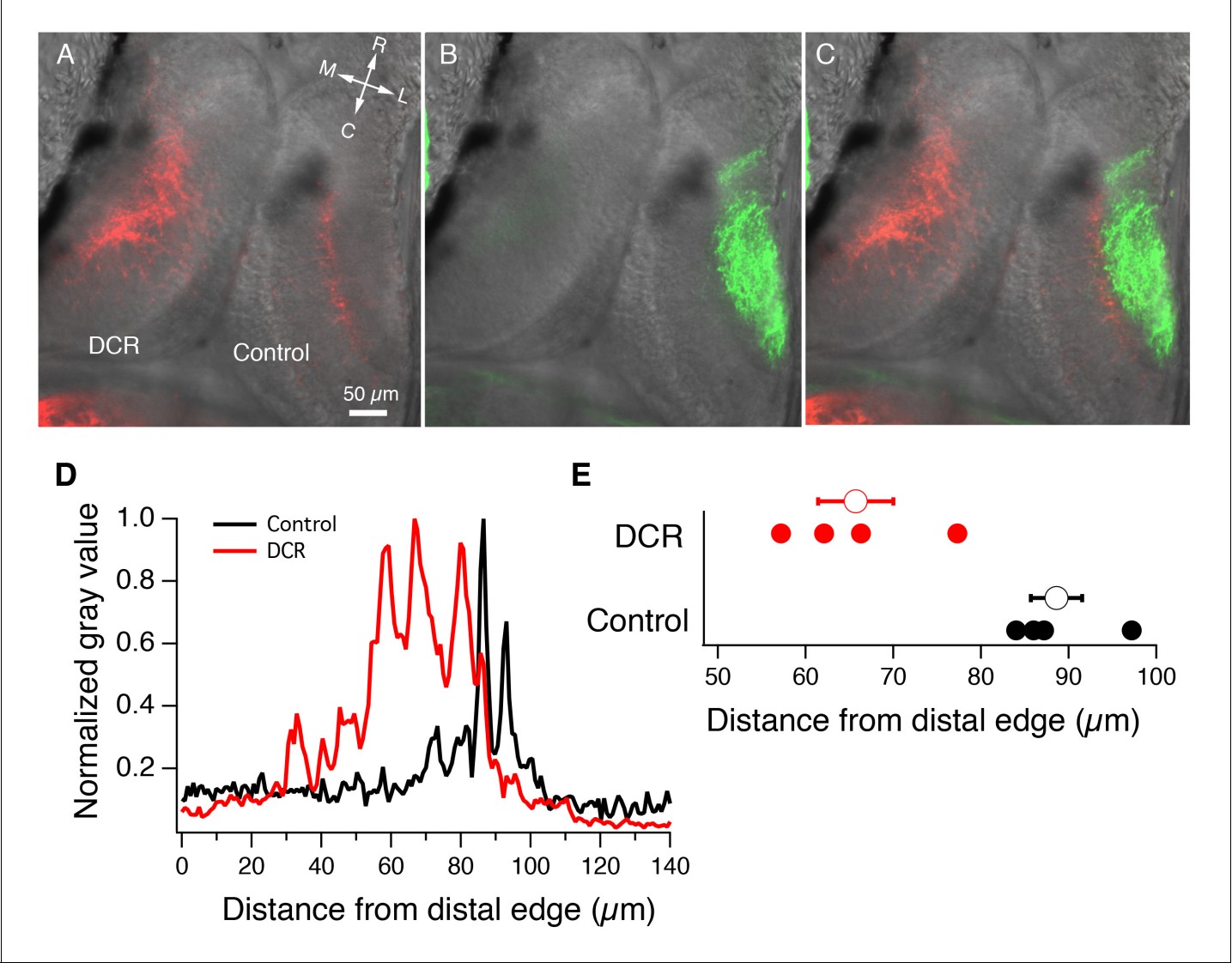

**Figure 4.** Monocular enucleation at stage 34 results in extended innervation of tectal neuropil by HB axons. (**A**) Bilateral DiI (red) labeling of HB inputs shows extensive HB axon innervation of the region of neuropil normally occupied by RGC axons in the DCR tectum. (**B**) DiD (green) labeling of the RGC inputs coming from the intact eye shows RGC axon innervation of control tectum. (**C**) Merged image of panels A and B. All fluorescent images are superimposed on transmitted light images to show relationship of multisensory axons relative to the tectum. R: rostral, C: caudal, L: lateral, M: medial. (**D**) Spatial fluorescence intensity plots for the HB axon innervation of control and DCR tectum shown in panel A. Fluorescence intensity is expressed as 8-bit gray value and has been normalized to optimize comparison of the relative profiles of control and DCR tecta. Notice that the profile for the HB axons of the DCR tectum is shifted distally compared to control. (**E**) A plot showing the distance from the distal edge at which HB fluorescence peaks, for individual control and DCR tecta. Each solid dot represents an individual tecta, and the larger, empty circle shows the corresponding average distance from distal edge at which maximum HB axon fluorescence for control and DCR is observed.

## NMDAR blockade disrupts segregation of visual and non-visual inputs across the laminar axis of the neuropil

It has been established that blocking NMDARs during the formation of the retinotectal projection prevents refinement of the topographic map (*Dong et al., 2009*; *Cline and Constantine-Paton, 1989*) and the formation of eye-specific stripes in the 3-eyed frog (*Cline et al., 1987*). To determine whether NMDAR blockade disrupts the subcellular targeting of RGC and HB input to the distal and proximal regions, respectively, of tectal neuron dendrites, tadpoles were reared in the presence of the non-competitive NMDAR blocker, MK-801 (25 µM). Adding this drug to the rearing solution

chronically inhibits NMDAR activity in tadpoles and it is well established that this drug at this concentration abolishes NMDAR-mediated visual responses in tectal neurons (*Ruthazer et al., 2003*; *Dong et al., 2009*), but does not appear to significantly alter RGC- or HB-evoked transmission (data not shown), consistent with the observation that both RGC and HB-evoked input received by tectal neurons is mediated largely via AMPA receptors (*Deeg et al., 2009*) and that the presence of MK801 had negligible effects on RGC-evoked responses in tectal neurons (*Ruthazer et al., 2003*) Exposure to the blocker began at developmental stage 39, when RGC axons have just reached the tectum, through stage 49, when the retinotectal circuitry has formed and has undergone some degree of topographic refinement. Once MK-801-exposed and batch-matched controls (i.e. tadpoles generated from the same mating and reared in regular Steinberg's solution) had reached stage 49, the spatial pattern of functional synapses formed by the two sensory inputs was measured by recording FPs from different points along the laminar axis in response to RGC and HB activation. The FPs and image plots from the MK-801-exposed tecta were compared to those of batch-matched controls. We found that the MK-801 treated tecta displayed a noticeably different spatial pattern of both RGC- and HB-evoked FP amplitudes (*Figure 5A,B*). In control tecta, the distal part of the laminar axis receives the majority of RGC input while the proximal part receives the majority of HB input (*Figure 5A*). For MK-801 tecta however, the spatial distribution is much more disperse and without an obvious separation between the two inputs (*Figure 5B*). Also for each tectum, the position across the axis where the absolute maximum RGC- and HB-evoked FP amplitude occurred was compared for controls and for MK-801 treated tecta. We found that for controls, the maximum RGC-evoked FP amplitudes occurred at 56.66 ± 7.41 μm from the distal edge, n = 15, 95% CI: upper limit = 72.6 μm, lower limit = 40.8 μm; and maximum HB-evoked amplitudes occurred at 102.73 ± 5.41 μm from the distal edge, n = 11, 95% CI: upper limit = 114.8 μm, lower limit = 90.7 μm; p=0.0007, non-parametric Mann-Whitney test (*Figure 5C*), thus a distance of approximately 45 μm separates the maximum responses of the two inputs. For MK-801-reared tecta, the maximum RGC-evoked FP amplitudes occurred at 80.48 ± 6.88 μm from the distal edge, n = 21, 95% CI: upper limit = 94.8 μm, lower limit = 66.1 μm; and maximum HB-evoked amplitudes occurred at 97.5 ± 7.4 μm from the distal edge, n = 12, 95% CI: upper limit = 113.8 μm, lower limit = 81.2 μm; p=0.122, unpaired t-test (*Figure 5C*), thus a distance of approximately 17 μm separating them. Moreover, blocking NMDARs produced an FP amplitude spatial profile that lacked an obvious peak. In other words, the absolute peak amplitude was surrounded by amplitudes of similar magnitude (*Figure 5B*). Because of this, we also measured the amount of overlap between the HB- and RGC- evoked FP spatial profiles and observed significantly more overlap between RGC and HB in the MK-801 exposed tecta compared to the control tecta (*Figure 5D*; overlap index for control: 3.23 ± 0.42, n = 12, 95% CI: upper limit = 4.2 μm, lower limit = 2.3 μm; MK801: 4.42 ± 0.29, n = 12, 95% CI: upper limit = 5.1 μm, lower limit = 3.8 μm; p=0.029, unpaired t-test). CSDs were also calculated for the two groups and the resulting image plots were consistent with the FP analysis: In control tecta, the main RGC-evoked current sink is localized to the distal region of the laminar axis (*Figure 6A*), the HB-evoked current sink to the proximal region (*Figure 6B*). In MK-801 reared tadpoles, however, the RGC- and HB-evoked current sinks appear more diffuse is space and in some cases sinks and sources appeared throughout the entire length of the laminar axis (*Figure 6C,D*), indicating a disorganized projection. Similar to the FPs, the position across the axis where the maximum RGC- and HB-evoked CSD sinks occurred revealed less separation between inputs in MK-801-treated group due to decreased segregation of both inputs (*Figure 6E*): For control, the maximum RGC-evoked CSD sinks occurred at an average of 50.5 ± 8.9 μm from distal the distal edge (n = 18 tecta; 95% Cl upper limit = 97.94 μm, Cl lower limit = 63.6 μm), and major HB-evoked CSD sinks occurred at an average of 80.8 ± 7.9 μm from the distal edge (n = 13 tecta; 95% Cl upper limit = 69.28 μm, Cl lower limit = 31.83 μm). The average distance between the major HB- and RGC-evoked CSD sinks was statistically significant (p=0.02, unpaired t-test). For MK-801 tecta, the maximum RGC-evoked CSD sinks occurred at an average of 71.7 ± 9.6 μm from the distal edge (n = 18 tecta, 95% CI upper limit = 92.01 μm, lower limit = 51.32 μm), and the major HB-evoked CSD sinks occurred at an average of 70 ± 8.4 μm from the distal edge (n = 11 tecta; 95% Cl upper limit = 88.76 μm, lower limit = 51.24 μm) suggesting essentially no gap, between the inputs. Indeed, the average distance between the major RGC and HB CSD sinks in the MK-801 treated group was not statistically significant (p=0.91, unpaired t-test). Thus, the majority of the RGC synaptic input in MK801-treated tecta was happening further away

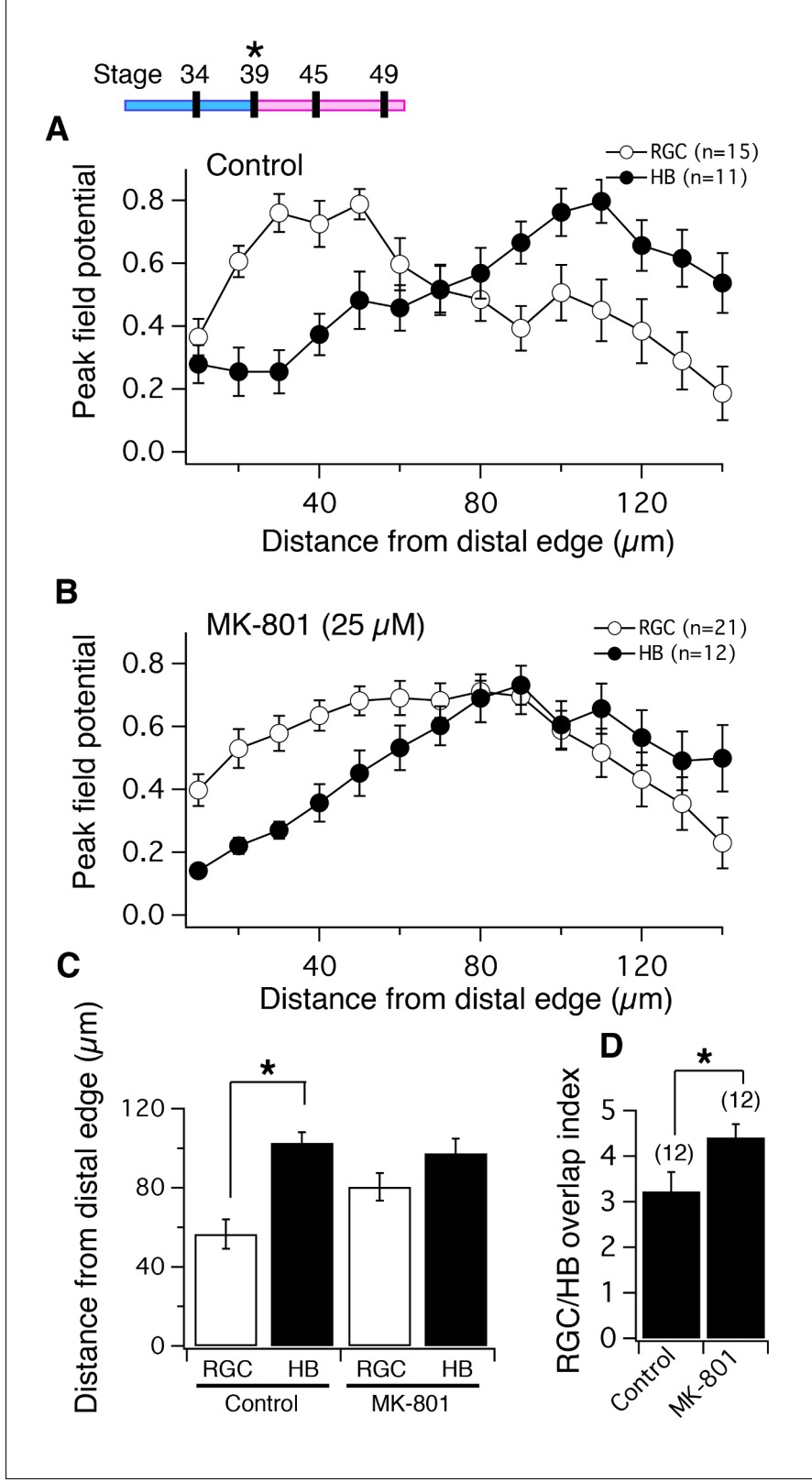

**Figure 5.** NMDA receptor blockade, starting at stage 39, disrupts the normal degree of spatial segregation between RGC and HB synaptic inputs measured by peak FPs. (*Top Left*) timeline of experiment. Average normalized peak FPs of RGC (white) and HB (black) inputs along the laminar axis for (**A**) control tadpoles and (**B**) tadpoles reared in MK-801. The spatial FP profiles in A show distinct and separated peaks formed by RGC and HB

*Figure 5 continued on next page*

*Figure 5 continued*

inputs, and this pattern is disrupted in MK-801-reared tadpoles. (**C**) Average distance from the distal edge at which the largest RGC- and HB-evoked FP amplitudes occurred in control and MK-801 tecta. Notice that this distance is significant in controls but not MK-801-reared tadpoles indicating less segregation. (**D**) RGC/HB axon overlap, expressed as the overlap index, in control and MK-801-reared tadpoles. There is significantly more overlap in the MK-801 group. Numbers above bars represent number of tecta.

from the distal edge compared to control, and the majority of the HB synaptic input was happening closer to the distal edge compared to control.

Chronic NMDAR blockade caused both sets of afferent inputs to shift closer to one another across the laminar axis, with a portion of the inputs moving outside the boundaries of their lamina and forming synapses where they normally would not. Overall, this resulted in a greater amount of spatial overlap of inputs at the border region. Besides appearing less confined to a specific lamina, the spatial pattern of synaptic connections displayed by both sets of inputs appeared less defined, lacking an obvious peak region of synaptic input.

## NMDAR blockade after the segregated pattern of inputs has been formed does not affect segregation of functional inputs

Having determined that normal segregation of RGC and HB inputs onto tectal neuron dendrites requires functional NMDARs, we next asked if the *maintenance* of this organization is also NMDAR-dependent. To address this, tadpoles were exposed to MK-801 at a later stage in development (stage 45), after both the visual and non-visual inputs have established their postsynaptic targets. Once tadpoles reached stage 49, the same set of experiments as previously described were carried out to characterize the degree of functional and structural segregation. The RGC- and HB-evoked spatial profile of peak FP amplitudes of the MK-801-exposed tecta resemble control profiles (*Figure 7A*), and we found that the maximum FP amplitude for both RGC and HB inputs occurred at approximately 40 and 100 µm, respectively, from the distal edge of the neuropil for both MK-801 and batch-matched control tecta (*Figure 7A*). Furthermore, the distance between the maximum RGC-evoked and HB-evoked FP amplitudes is significant for both MK-801 (*Figure 7B*; RGC distance = 46±2.44 µm, 95% CI: upper limit = 52.8 µm, lower limit = 39.2 µm; HB distance = 100±3.16 µm, n = 5; 95% CI: upper limit = 108.8 µm, lower limit = 91.2 µm; p=0.01, non-parametric Mann-Whitney test) and control tecta (*Figure 7B*; RGC distance = 53.33±13.33 µm, 95% CI: upper limit = 110.7 µm, lower limit = −4 µm; HB distance = 110±7.07 µm; n = 3, 95% CI: upper limit = 132.5 µm, lower limit = 87.5 µm; p=0.047, non-parametric Mann-Whitney test), indicating that NMDAR activation at stage 45 is not required for the proper segregation of multisensory axons along the laminar axis of the neuropil. Analysis of CSDs reveals that, similar to control tecta, the main RGC-evoked sinks were localized to the distal portion of the neuropil (*Figure 7C*), the HB sinks to the proximal portion, in tadpoles exposed to MK-801 later in development (*Figure 7D*). Statistical analysis of CSDs indicated that the distance between the major RGC- and HB-evoked sink was significant for both control tecta and the tecta of tadpoles that had been exposed to MK-801 between stages 45 and 49 (*Figure 7E*). For control tecta, the major RGC-evoked CSD sink occurred at an average of 26.6 ± 16.6 µm from distal the distal edge (n = 3 tecta; 95% CI upper limit = 98.38 µm, CI lower limit = 45.04 µm), and the major HB-evoked CSD sink occurred at an average of 92.5 ± 11.1 µm from the distal edge (n = 4 tecta; 95% CI upper limit = 127.8 µm, CI lower limit = 57.22 µm). The average distance between the major HB- and RGC-evoked CSD sinks was statistically significant (p=0.018, unpaired t-test). For tadpoles exposed to MK-801 between stage 45 and 49, the maximum RGC-evoked CSD sinks occurred at an average of 40 ± 5.5 µm from distal the distal edge (n = 5 tecta, 95% CI upper limit = 55.21 µm, lower limit = 24.79 µm), and the major HB-evoked CSD sinks occurred at an average of 74 ± 5.1 µm from the distal edge (n = 5 tecta; 95% CI upper limit = 88.2 µm, lower limit = 59.84 µm). This distance between the major RGC- and HB-evoked CSDs was statistically significant (p=0.002, unpaired t-test).

To determine whether the effects of early (stage 39) versus late (stage 45) NMDAR blockade on the spatial pattern of functional synaptic input made by the RGC and HB inputs (quantified via FP and CSD analysis) is mirrored by the pattern of axonal terminations across the neuropil, DiD (green)

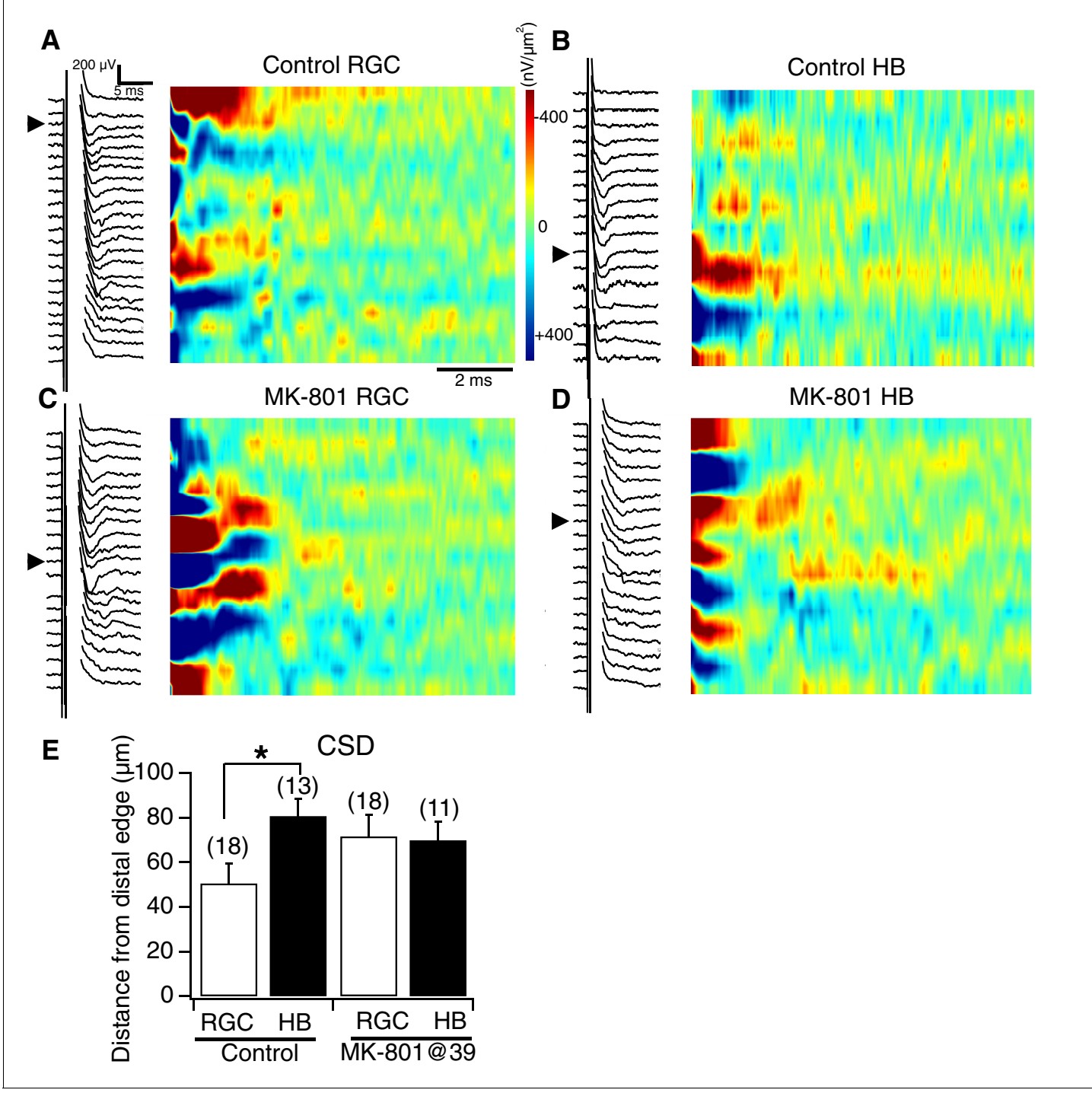

**Figure 6.** NMDA receptor blockade, starting at stage 39, disrupts the normal degree of spatial segregation between RGC and HB synaptic inputs: sample image plots. (**A**) Control RGC-evoked FPs with corresponding image plot. Main RGC-evoked sink is localized to the most distal end of the neuropil, with an additional minor sink in the proximal region. (**B**) Control HB-evoked FPs with corresponding image plot. Main HB sink is localized to the proximal region of the tectal neuropil. (**C**) MK-801 RGC-evoked FPs with corresponding image plot. Main RGC-evoked sink now appears in the proximal region of the neuropil instead of the distal region. (**D**) MK-801 HB-evoked FPs with corresponding image plot. Main HB-evoked sink now appears in both the distal region and proximal region of the tectal neuropil. Arrowheads refer to the site of the largest peak FP. (**E**) Bar graph summarizing the location of major RGC- and HB- evoked CSD sinks along the distal-proximal axis. For control tecta, the distance between the major RGC- and HB-evoked CSD sinks is significant, but not for tecta exposed to MK801 at stage 39.

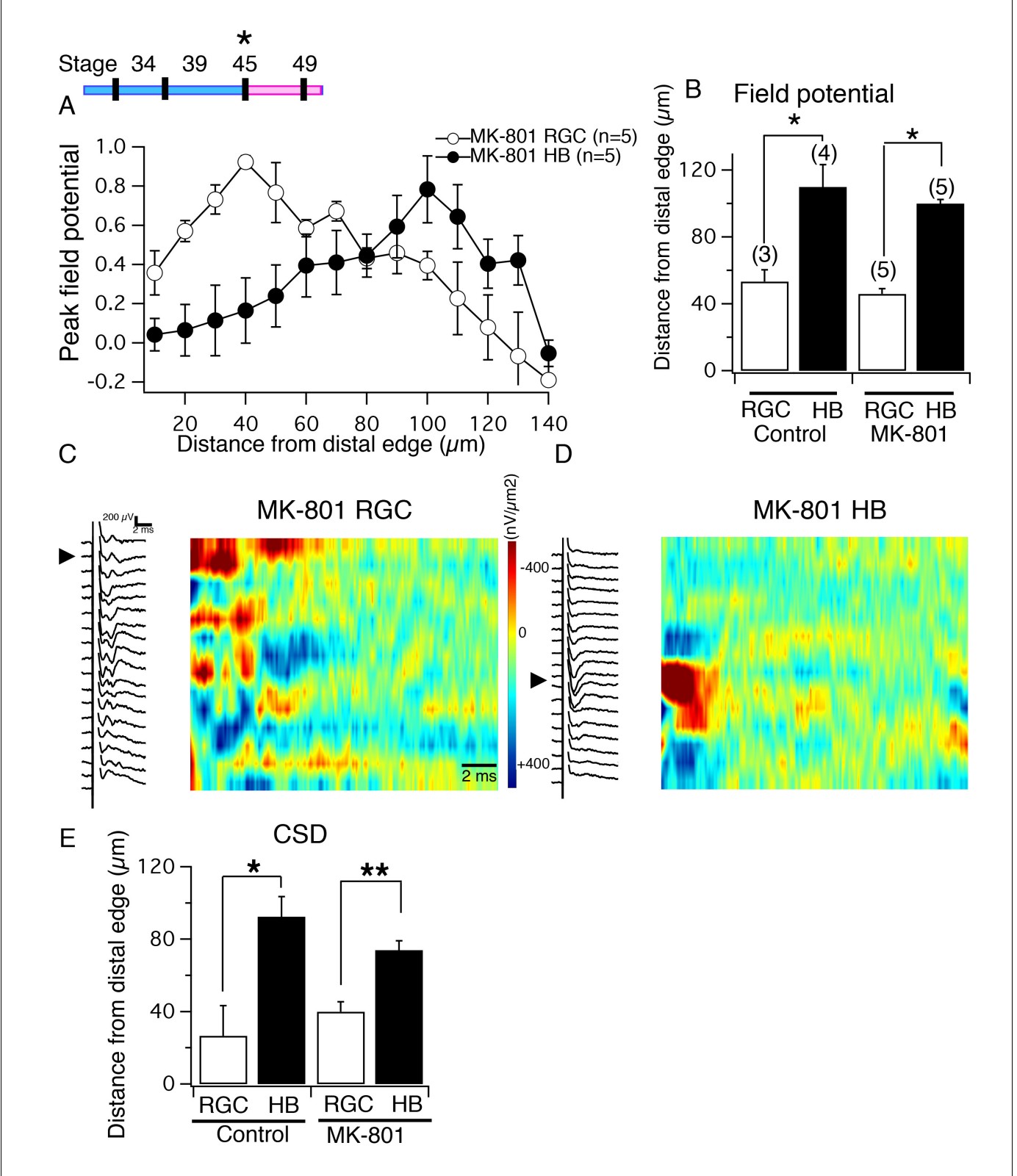

**Figure 7.** NMDA receptor blockade, starting at stage 45, does not disrupt the normal degree of spatial segregation between RGC and HB synaptic inputs. (*Top, left*) Timeline of experiment. (**A**) Average normalized peak FPs of RGC-evoked (white) and HB-evoked (black) inputs along the laminar axis of MK-801-reared tadpoles. The spatial FP profiles show distinct and separated peaks formed by RGC and HB inputs. (**B**) Average distance from the distal edge at which the largest RGC-evoked and HB-evoked FP amplitudes occurred in control and MK-801-reared tecta. (**C**) RGC-evoked FPs with
*Figure 7 continued on next page*

Figure 7 continued

corresponding image plot showing the largest RGC-evoked sink localized to the most distal region of the laminar axis. (D) HB-evoked FPs with corresponding image plot showing the largest HB-evoked sink localized to the proximal region of the laminar axis. (E) Bar graph summarizing the location of major RGC- and HB- evoked CSD sinks along the distal-proximal axis. For both control tecta and tecta exposed to MK-801 at stage 45, the distance separating the major RGC- and HB-evoked CSD sinks is significant.

and DiI (red) was injected into the eyes and HBs, respectively, of control and MK-801-reared tadpoles to visualize RGC and HB axon terminals in the neuropil. *Figure 8* shows examples of a control, and an early (stage 39), and late (stage 45) NMDA blockade tectum with fluorescently labeled RGC and HB axons, and the corresponding spatial fluorescent intensity profile for each. Compared to control, blocking NMDARs early during circuit development (stage 39) shortened the average distance between peak RGC and HB axon fluorescence (*Figure 8A,B and D*; distance between peak fluorescence for RGC and HB axons for control tecta: $57.1 \pm 4.0$ μm, n = 5, 95% CI upper limit = 65.58 μm, CI lower limit = 52.62 μm; for early MK-801 tecta: $40.45 \pm 2.15$ μm, n = 11 tecta; 95% CI upper limit = 47.23 μm, CI lower limit = 34.54 μm; p=0.001 compared to control distance; unpaired t-test), while blocking NMDARs later in development (stage 45) did not appear to elicit a significant effect on this distance (*Figure 8C and D*; late MK-801 tecta: $50.8 \pm 7.0$ μm, n = 5, 95% CI upper limit = 70.18 μm, CI lower limit = 31.42 μm; p=0.45 compared to control distance, unpaired t-test). The effect of early but not late NMDAR blockade on the distance between peak axon fluorescence mirrors the observed effect on the position of peak FPs and CSD sinks and suggests that the location of the peak axon fluorescence is also the location of the major synaptic input.

In addition to measuring the distance between the peak fluorescence of RGC and HB axons, the amount of overlap between inputs was also calculated by measuring the area of neuropil that was common to both inputs. We found that early and late NMDAR blockade significantly increased the total amount of overlap of RGC and HB axons compared to control (*Figure 8A–C and E*; the average RGC and HB overlap area for control: $12.5 \pm 1.7$ gray value * μm, n = 7, 95% CI upper limit = 15.82, CI lower limit = 9.19 gray value * μm; for early MK-801 tecta: $23.1 \pm 3.7$ gray value * μm, n = 11, 95% CI upper limit = 30.3, CI lower limit = 15.9 gray value * μm; p=0.04 compared to control, un-paired t-test; for late MK-801 tecta: $29.85 \pm 4.4$ gray value * μm, n = 5, 95% CI upper limit = 38.4, CI lower limit = 21.3 gray value * μm; p=0.002 compared to control). Thus, although blocking NMDARs later in development did not disrupt the distance between peak FPS, CSDs, or axon fluorescence, it did appear to activate a fraction of axons – perhaps a set of relatively immature axons that are still undergoing NMDAR-dependent processes – to desegregate. This effect is clearly illustrated in the fluorescent profile examples shown in *Figure 8*. Compared to the fluorescent profile for the control tectum (*Figure 8A*, far right), the profile for the late MK801 tectum (*Figure 8C*, far right) shows a noticeable increase in overlap of the two inputs. In this example, the overlap appears to be generated mainly by a set of low intensity (diffuse) HB axons that have moved into the distal territory. The overall result is increased overlap between inputs without altering the position of the peak axon fluorescence.

Combined, these data suggest that, once formed and likely stabilized via NMDAR-dependent mechanisms, sensory-specific axon segregation is no longer NMDAR-dependent for the majority of axons. Thus, there appears to be a critical period or window in developmental time in which the inputs will shift across the laminar axis in the absence of NMDAR activity.

## Discussion

The major findings of this study are (1) During the development of the *Xenopus* tadpole tectal circuitry, absence of RGC axons in the optic tectum causes the mechanosensory inputs to shift distally along the laminar axis of the neuropil such that a fraction of these inputs move into the neighboring vacant lamina and form synaptic connections with distal dendrites that are normally dendritic region innervated by RGC inputs. (2) Similarly, chronic blockade of NMDA receptors during circuit development desegregates the two different sets of sensory inputs, showing that the subcellular targeting of the different afferent inputs to specific regions of a common tectal dendrite requires NMDAR activation (*Figure 9*). (3) Blocking NMDARs after the circuit has formed fails to disrupt the pattern of

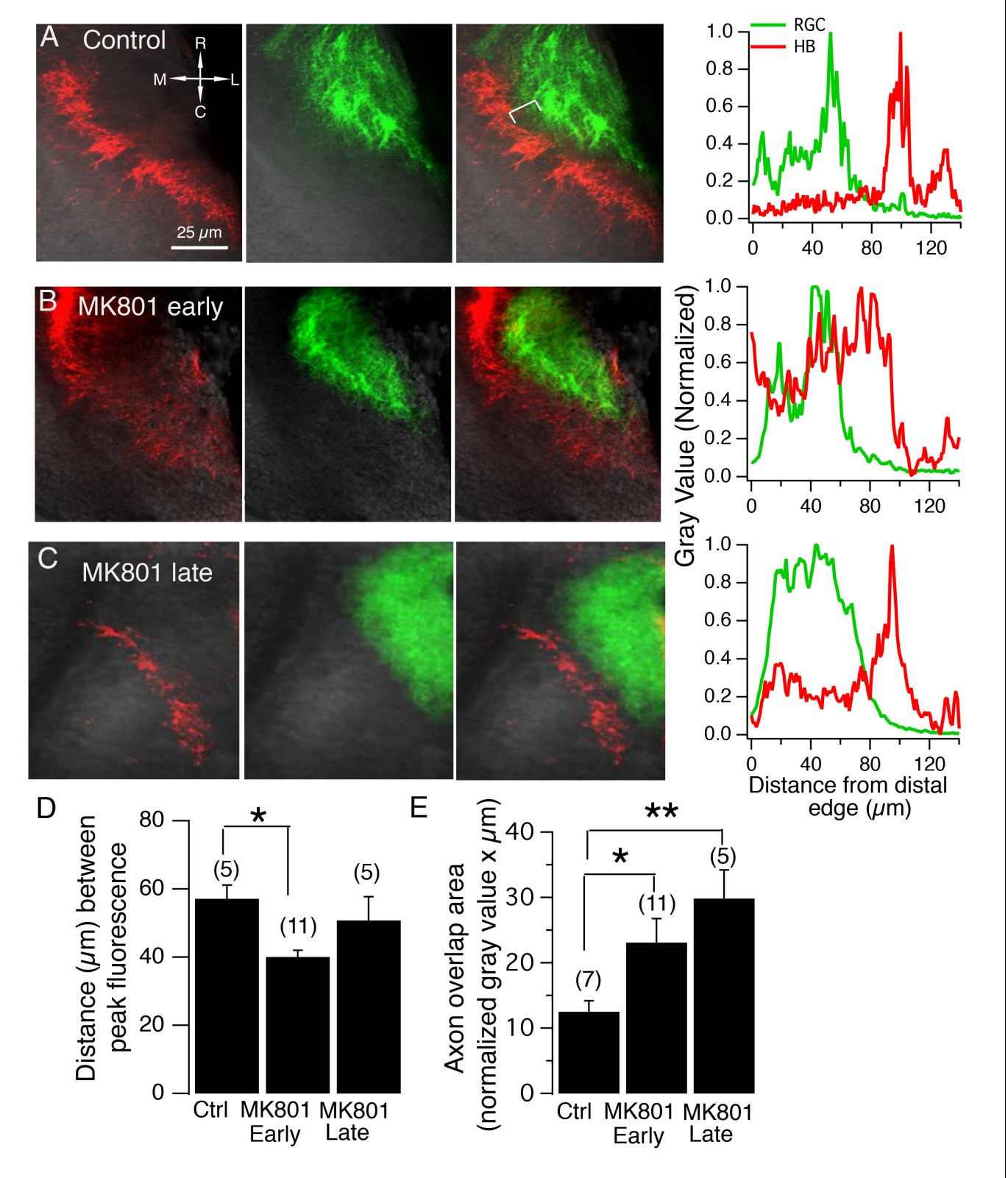

**Figure 8.** NMDA receptor blockade, starting at stage 39, but not stage 45, disrupt the normal degree of spatial segregation between RGC and HB axons. (A) Control HB afferent input (red) projects to the proximal (medial) region of the tectal neuropil, RGC afferent input (green) projects to the distal (lateral) region of the tectal neuropil. In the merged image, notice the gap, indicated by the white bracket, between two sets of sensory input, and the corresponding gap in peak axon fluorescence shown in the corresponding fluorescent intensity profile (*far right*). (B) HB and RGC axons innervating the
*Figure 8 continued on next page*

*Figure 8 continued*

tectum of a tadpole exposed to MK801 at stage 39 ('early') and corresponding fluorescent profile (*far right*). There appears to be a less distinct gap between the two inputs, suggesting less segregation. (**C**) HB and RGC axons of a tadpole exposed to MK801 at stage 45 ('late'), and corresponding fluorescent profile (*far right*). The merged image and fluorescent profile suggest that MK-801 rearing at stage 45 does not disrupt the segregation along the tectal neuropil at stage 49. Notice the gap between the two sets of input. (**D**) Bar graph summarizing the distance between peak HB and RGC axon fluorescence. (**E**) Bar graph summarizing the amount of overlap between HB and RGC axons. R: rostral, C: caudal, L: lateral, M: medial.

functional synaptic inputs, indicating a critical period for the sculpting of lamina and the subcellular pattern of targeting of axonal inputs.

We observed that removing the contralateral RGC inputs before they reached the tectum caused the entire spatial pattern of mechanosensory (HB) synaptic input to shift distally. A portion of the HB inputs extended beyond their normal proximal lamina into a more distal lamina, and formed synapses onto more distal regions of dendrite that are normally synaptic targets for RGC axons, suggesting that the sensory-dependent segregation of afferent inputs across the neuropil, and in this case, across the distal-proximal axis of individual dendrites, requires their physical presence. If the input is absent during the development of the circuit, its target region becomes ectopically innervated by the remaining neighboring afferent input. Similar monocular enucleation studies in the axolotl (*Harris, 1983*) and the rodent (*Rhoades, 1980*; *Rhoades et al., 1981*) are consistent with our findings. In both of these studies, eye enucleation during embryonic stages of development resulted in a portion of the non-visual inputs to shift into the vacant area of the optic tectum for the axolotl, and the superior colliculus in the rodent. In cats and primates monocular enucleation leads to the inputs from the remaining eye to grow beyond their normal laminae in the lateral geniculate nucleus and take over territory that normally would have been occupied by the enucleated eye (*Rakic, 1981*; *Chalupa and*

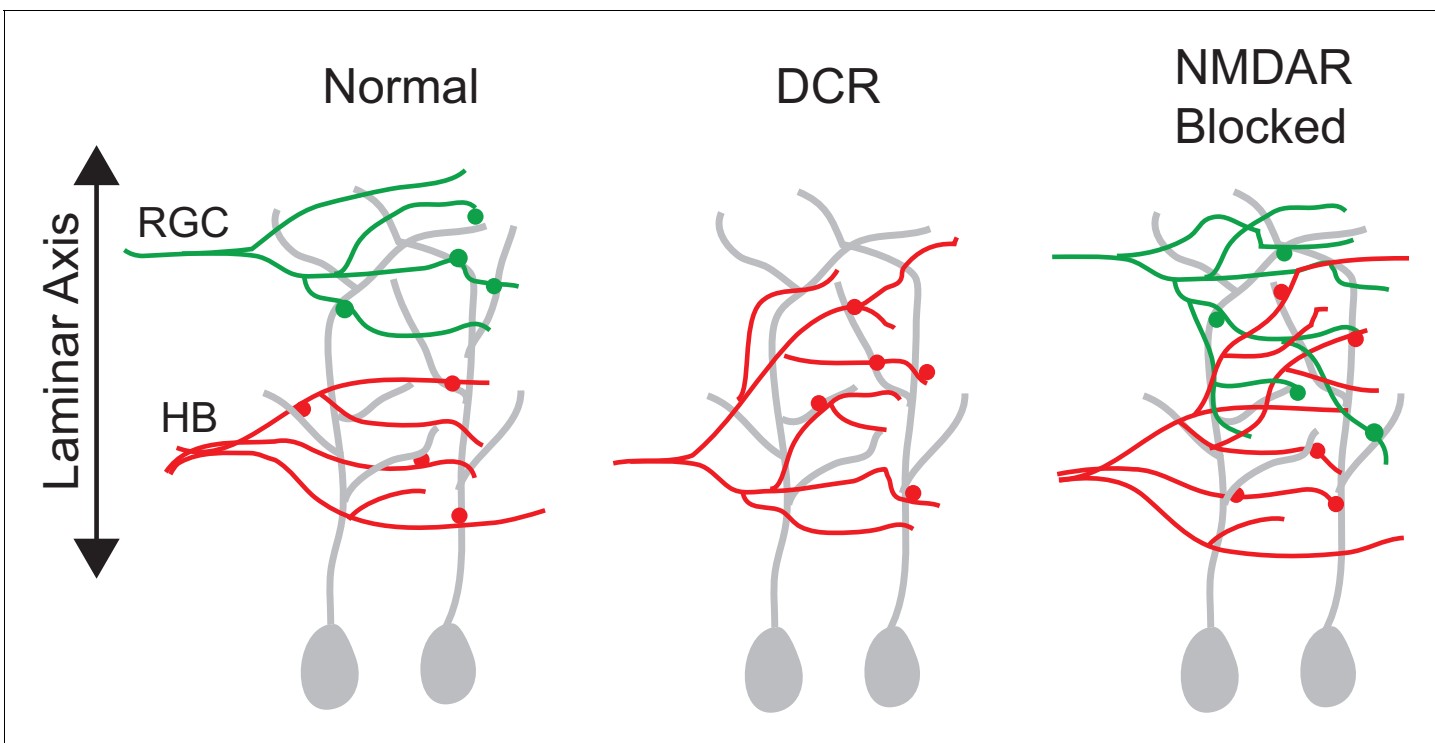

**Figure 9.** A summary figure showing the effect of monocular enucleation and NMDAR blockade at stage 39. RGC and HB afferent inputs arrive to the tectum at the same time (stage 39) and remain segregated along the laminar axis. Monocular enucleation before as RGC axons are just exiting the eye (stage 34) results in HB axons extending into tectal dendritic territory normally occupied by RGC axons. NMDAR blockade during synapse formation (stage 39), but not after synapses have been stabilized (stage 45), results in the desegregation of RGC and HB axons in the laminar axis.

*Williams, 1984*; *Shatz, 1996*; *Shatz and Sretavan, 1986*). Thus, several pieces of evidence from different model systems support the concept that a reciprocal interaction between different sensory inputs shapes their segregation. From a comparative standpoint, it would be interesting to know if this type of enucleation-induced remodeling of inputs happens in the zebrafish or chick optic tectum, because laminar segregation in these models is thought to be mediated by activity-independent molecular cues, namely cell adhesion molecules (*Baier, 2013*; *Xiao and Baier, 2007*; *Yamagata and Sanes, 1995*; *Yamagata et al., 1995*; *Takahashi et al., 1999*), however, there are no reports of enucleation studies in these models.

The monocular enucleation experiment data also provide evidence that in the tadpole tectum, the sensory-dependent subcellular targeting of axons to specific regions of a common dendrite is likely not guided by guidance molecules, such as ephrins, alone because if it were, then just removing the RGC input would not be expected to desegregate the remaining input. A rewiring study by Lyckman and colleagues (*Lyckman et al., 2001*) provides an example of what we may have expected to find if the segregation of sensory inputs in the tadpole optic tectum were regulated by a molecular signal: in the mouse thalamus, RGCs project to the lateral geniculate nucleus (LGN) and auditory inputs project to the medial geniculate nucleus (MGN). It was found that cutting the auditory afferent input induced the RGC inputs to shift over to the MGN *only* in an EphrinA2/A5 double knockout mouse. The shift was not observed in wild type mice because Ephrin expression between the LGN and MGN inhibits the remaining input from crossing over into vacant territory. Therefore, because we observe a shift in mechanosensory inputs to the more distant dendrites by just removing the RGC input indicates that, in this case, segregation is not under the sole control of molecular cues. Nevertheless, the possibility that the absence of the RGC input somehow altered the expression pattern of some putative molecular signal cannot be ruled out, and it is interesting to note that a differential pattern of ephrin expression across the laminar axis of both the Rana pipiens (*Scalia and Feldheim, 2005*) and Xenopus laevis (*Higenell et al., 2012*) optic tectum has been reported, although whether this gradient is involved with segregation of the different sensory inputs has not been determined.

## Segregation of visual and non-visual inputs is likely achieved via an NMDAR correlation-based mechanism

The second main finding is that blocking NMDARs with MK-801 during the development of these circuits also disrupts the normal degree of segregation between inputs. These data suggest that the subcellular segregation of inputs across tectal neuron dendrites could be governed by an NMDAR-dependent correlation-based mechanism similar to the mechanism underlying refinement of the topographic map (*Cline and Constantine-Paton, 1989*; *Debski et al., 1990*; *Dong et al., 2009*). Our results from both the eye enucleation and NMDAR blockade experiments support a Hebbian form of developmental plasticity that dictates that inputs are governed according to the degree of correlation that exists between their firing patterns and the firing patterns of the local neighboring inputs. If a given input is well correlated with that of surrounding inputs, there is a high probability it will be stabilized. Conversely, if there is a low degree of correlation, it will be eliminated (*Ruthazer et al., 2003*; *Munz et al., 2014*). This concept of correlated inputs wiring to common regions of dendrite is supported by the finding that RGC inputs form a subcellular map across individual tectal neurons such that near-neighbor RGC inputs, which would display a high degree of correlated firing, target to near neighbor region of dendrite (*Bollmann and Engert, 2009*), and further the super-correlated pattern of RGC firing created by optic flow across the retina has been shown to enhance refinement (*Hiramoto and Cline, 2014*). Thus, across a common dendrite, any HB axon that is in the vicinity of RGC axons would not be able to form stable synapses in that area because the firing patterns of HB inputs are not correlated with RGC patterns. And because weak synapses do not support the stabilization of its axon (*Ruthazer et al., 2006*) – the entire HB axon and its synapses would be eliminated. Moreover, the normal distance, or gap, between the RGC and HB inputs could be a direct reflection of dendritic territory over which NMDARs can detect two given inputs as being correlated or non-correlated. Our results are consistent with NMDARs working as correlation detectors, strengthening what is perceived by the postsynaptic neuron as strong highly correlated input. Exposing tadpoles to MK-801, however, blocks all NMDARs, so the possibility that presynaptic NMDARs could play a role in axon segregation cannot be ruled out. It is not possible to block specifically postsynaptic NMDARs in essentially all tectal neurons that these tectum-wide

experiments require, and this is a limit of our preparation. Alternatively, a postsynaptic NMDAR-dependent mechanism would likely involve the downstream activation of calcium-calmodulin-dependent protein kinase II (CaMKII), which can be blocked pharmacologically. Thus, a future experiment is to test whether blocking CaMKII at developmental stage 39 disrupts segregation. This will help to determine presynaptic versus postsynaptic NMDAR-dependent mechanisms. In addition, this will be an especially interesting experiment given that postsynaptic activation of CaMKII in cultured cortical neurons has been shown to remodel inputs such that the strongest inputs are kept and strengthened while other inputs are completely lost (*Pratt et al., 2003*).

We observed that blocking NMDARs during circuit formation did not induce a global and complete remodeling of axonal inputs. Indeed, the inputs retained their normal patterns of innervation, and were targeted to their proper regions – RGC inputs to the distal region of the laminar axis, HB inputs to the proximal. Thus, similar to the formation of the topographic map, the guidance and targeting of different sensory inputs is almost certainly achieved by activity-independent molecular cues. But our data also indicate that, similar to the activity-dependent refinement phase of topographic map formation, the normal degree of segregation of inputs at the more local, synaptic level requires NMDAR-dependent activity. It is worth noting here that the segregation of different sensory inputs differs from refinement of the topographic map in that the segregation of inputs does not refine, technically. Instead of starting out coarse like the initial topographic map, meaning that the axons are less focused in space such that one postsynaptic neuron receives input from many axons and being sculpted via activity such that any one neuron receives fewer different axonal inputs, the different sensory inputs appear segregated ab initio, from the very beginning (*Deeg et al., 2009*; *Hiramoto and Cline, 2009*). Thus, it is not that the inputs start out overlapping and then become refined by activity, but blocking NMDAR activation while the sensory inputs are in the midst of forming synapses with postsynaptic targets causes the inputs to lose their normal degree of segregation. In this context, therefore, NMDAR-dependent activity prevents the desegregation of inputs.

Similar to the remodeling of axonal inputs underlying ocular dominance plasticity, the desegregation of inputs induced by blocking NMDARs also appears to be defined by a critical period. The critical period appears to coincide with when the sensory inputs are forming synaptic connections with postsynaptic tectal neurons. Once this dynamic phase of synapse formation has waned, blocking NMDARs no longer has an effect on the degree of segregation between the majority of afferent axonal inputs. It is interesting to note that although the late (stage 45) MK-801 exposure was not observed to alter the spatial position of peak field potential amplitudes, peak CSD sinks, nor peak HB and RGC axon fluorescence, it did appear to significantly increase the overall amount of overlap between these two sets of axons. This increase in overlap in response to late MK801 exposure could reflect a subpopulation of relatively immature axons that have more recently entered the tectum and are still undergoing NMDAR-dependent correlation-based processes. In addition, the HB axons, specifically, appeared to be more affected by late MK-801 exposure than the RGC axons (*Figure 8C*, far right). This is consistent with the report that synapses made by non-visual mechanosensory inputs mature slower and display lower AMPA:NMDA ratios compared to retinotectal synapses (*Deeg et al., 2009*).

Why may the segregation of inputs involve an activity-dependent component while many other examples of lamination appear to be hard wired? One important aspect of the tadpole optic tectum that sets it apart from most other integration centers is that the tectal neuron dendrites span the entire laminar axis and receive input from both sensory modalities. Thus, in the tadpole optic tectum, inputs are segregated to the distal and proximal regions of a common dendrite, while in zebrafish, and chick, inputs form onto specific cell types (*Yamagata and Sanes, 1995*; *Baier, 2013*). The subcellular type of targeting of inputs that occurs in the tadpole optic tectum also occurs in the hippocampus where individual pyramidal neurons receive two distinct, spatially segregated sets of afferent input: entorhinal cortical input which innervates the distal region, and local commissural input which is targeted to the more proximal region of the same dendrite (*Supèr and Soriano, 1994*; *Supèr et al., 1998*; *del Río et al., 1997*). The way inputs get targeted to different regions of common dendrite in these neurons is elaborate. Early on, there are different cell types (Cajal-Retzius and GABAergic cells) residing in two different laminae – and these cells are transient and they seem to attract the different inputs – once the inputs are there they actually form synapses with these transient populations of cells, then the attractor cells die off and the axons are left to form synapses with the local dendrites of the pyramidal cells. So it is difficult to say if this shares any mechanisms with

tectal neurons, except perhaps the first phase of segregation that involves the Cajal-Retzius and GABAergic attractor cells is independent of activity, but then NMDAR-dependent mechanisms maintain segregation of inputs as they form synapses onto the pyramidal dendrites.

### Possible function of subcellular segregation

This study does not address the function of subcellular segregation of inputs. We have not yet been able to address this because the chronic blockade of NMDARs – which disrupts the segregation – also disrupts the refinement of the RGC topographic map (*Dong et al., 2009*). In order to begin to understand why segregation of inputs across the laminar axis may be important, it will first be necessary to find a manipulation that disrupts specifically the segregation. The following ideas about possible functions of such segregation are speculative. Segregation during the formation of individual topographic maps may be conducive for the formation and refinement of these maps, because if there was no such segregation, then visual and non-visual input could potentially compete for the same postsynaptic sites and thereby interfere with necessary competition-based mechanisms which underlie refinement. Further, the segregation of different sensory inputs along the laminar axis could be necessary for the proper alignment of their topographic maps via promoting cooperation between different sensory inputs whose receptive fields match. If two axons – one at the distal end of the dendrite, one at the proximal end – contribute to generating an action potential in the postsynaptic neuron, then they will be kept and strengthened. This function of segregation is based on the assumption that the different sensory inputs are, at least a portion of the time, coactive. This subcellular cooperation between different sensory inputs would likely be weakened or made less robust if these two inputs were competing for the same region of dendrite. Thus, keeping the different inputs separated in space, yet still synapsing onto the same neuron could facilitate the alignment of the maps and thereby support proper multisensory integration. The spatial segregation of inputs also focuses or clusters correlated synaptic input onto a defined region of the dendritic tree. This clustering could promote cooperation between correlated inputs which may lead to long-lasting changes in synaptic strength (*Frick et al., 2004*) and dendritic excitability (*Losonczy et al., 2008*). Clustered input, as opposed to a disperse pattern, is thought to potentiate spike-timing plasticity processes (*Sjostrom et al., 2008*) which are expressed in these neurons (*Zhang et al., 1998*).

### Conclusion

During development of many sensory circuits in the vertebrate CNS, molecular cues guide sensory inputs to form a course topographic map across the 2-dimensional surface of their target regions. The course topographic map is refined via NMDAR-dependent fire-together-wire-together rules (*Cline, 1998*). Our results suggest a similar progression of activity-independent, then –dependent mechanisms that together result in the segregation of inputs across tectal neuron dendrites. Activity – specifically NMDAR-dependent activity – segregates the different inputs, implicating a correlation-based mechanism. Inputs that fire together, do wire to common regions of dendrite, and inputs that don't, won't.

## Materials and methods

All experimental protocols have been approved by the University of Wyoming's Institutional Animal Care and Use Committee (IACUC). Wildtype *Xenopus laevis* tadpoles were raised in Steinberg's solution at 25°C on a 12:12 hr light-dark cycle, and developmental stages were identified according to *Nieuwkoop and Faber (1994)*. All experiments were carried out using a modified whole brain preparation as described in *Hamodi and Pratt (2015)*. Briefly, tadpoles were anesthetized in 0.01% MS-222 and pinned to a block of silicone elastomer submerged in external recording saline (in mM: 115 NaCl, 2 KCl, 3 CaCl$_2$, 3 MgCl$_2$, 5 HEPES, and 10 glucose, pH 7.25, osmolarity 255 mosM). Next, the skin overlying the brain was peeled away and the brain and brainstem were filleted open along the dorsal midline, and the most lateral 1/4th of the tectum (corresponding to the most dorsal 1/4th in vivo) is excised. Next, the preparation is pinned onto the side of a piece of submerged silicone, positioned such that the sliced side is on the surface and readily accessible for recording, and also so that bipolar stimulation electrodes can be placed onto the optic chiasm and hindbrain to stimulate the visual (RGC) and non-visual (HB) inputs, respectively (*Hamodi and Pratt, 2015*).

Typically, 3–5 tecta per experimental group were recorded from a given batch/clutch of tadpoles. All recorded tecta were analyzed and none was excluded from the dataset.

## Experimental manipulations

### Monocular enucleation

Monocular enucleations were done at stage 34 using a sterile 26-gauge syringe needle. Tadpoles were first anesthetized using 0.02% MS-222 and the entire eye, from lens to choroid, was removed. The animals recovered from the operation in Steinberg's solution. Post-surgery, tadpoles displayed normal swimming and schooling behaviors, similar to controls. Electrophysiological experiments and axonal dye fills were performed approximately ten days after the operation.

### Chronic blockade of NMDA receptors

To chronically block NMDA receptors, tadpoles were reared in Steinberg's solution containing 25 µM (+)-MK-801 (Tocris) starting at stage 39 for 8–10 days, and the drug-containing media was refreshed every three days (*Dong et al., 2009*). Electrophysiological recordings and axonal dye fills were conducted at stage 49. All recordings were carried out in the absence of MK801 to assess the effects of chronic NMDAR blockade, not acute. To promote washout of MK801, tadpoles were anaesthetized and dissections carried out in the absence of the drug, a combined period of approximately 30 min prior to recording.

## Whole-cell electrophysiology

Tectal neurons were visualized using a Zeiss light microscope equipped with a 60X water-immersion objective and a Hamamatsu infrared charge-coupled device camera. For whole-cell recordings, we used glass micropipettes with 8–12 MΩ resistance filled with $K^+$-gluconate internal recording saline (in mM: 100 K-gluconate, 8 KCl, 5 NaCl, 1.5 MgCl2, 20 HEPES, 10 EGTA, 2 ATP, and 0.3 GTP, pH 7.2; osmolarity 255 mosM). Electrophysiological recordings were carried out using an Axon Instruments 700B Multipatch amplifier (Molecular Devices, Sunnyvale, CA), digitized at 10 kHz using a Digidata 1322A digitizer, and recorded using pCLAMP software.

## Extracellular field-potential recordings

All field potential (FP) recordings were done as described in *Hamodi and Pratt (2015)*. Briefly, we used glass micropipettes in the range of 4–5 MΩ filled with external recording saline. In current clamp mode, evoked FPs in response to stimulating either the RGC (visual) or hindbrain (non-visual) inputs were recorded at 10 µm intervals from the most outer/distal part of the neuropil to the deepest somatic layer while stimulating RGC or HB axons. A scale bar was superimposed onto the video monitor for accurate placement of the recording pipette. The strength of stimulation was determined by first identifying the intensity that generates a maximal FP response then decreasing that intensity to about 75% of maximum to generate a reliable synaptic FP while minimizing action potential firing and thereby minimizing any depolarizations in the dendrites induced by back-propagating action potentials. Importantly, the modified brain slice preparation used here allows for both visual and non-visual mechanosensory inputs to be activated and the resulting synaptic field potentials to be recorded from readily accessible and accurately measured positions along the entire laminar axis of the neuropil (*Hamodi and Pratt, 2015*). Unlike recording from the traditional whole brain preparation, the recording pipette is never advanced vertically through the tissue, which creates unidentified amounts of drag and thus inaccurate distance measurements and has prevented the recording of the relatively small and thin neuropil of tadpole larvae (*Chung et al., 1974*). With the modified preparation used here, the recording pipette is placed in the tissue at a particular point along the laminar axis to measure evoked FPs, then raised up off the tissue, moved to the next successive point along the laminar axis and lowered back down into the tissue for the next recording.

## Quantification of field potential amplitudes

FP amplitudes reflect the peak deflection that occurred within 4 ms of the time of stimulus since the monosynaptic response is known to occur within this time frame (*Pratt and Aizenman, 2007*). Peak FP amplitudes of RGC and HB responses varied across tecta, therefore we normalized these responses such that the largest amplitude observed for each individual tectum was designated as

1.0. In this way, responses across tecta could be averaged. We report the distance at which FP peak amplitudes occur as the distance from distal edge because this edge is readily identified and therefore provides a precise point in space from which to measure all other points across the laminar axis. For each tectum, a spatial profile of functional RGC and HB-evoked synaptic input was generated by plotting the normalized peak FP amplitude as a function of position across the distal-proximal axis. Besides showing where along the axis the bulk of the synaptic input resides, these plots were also used to measure the degree of spatial overlap between the two sensory inputs by calculating the shared region of the plot (i.e. the portion of synaptic profiles that are common to both inputs). The area of the shared region was estimated by summing all the portions of the field potential amplitudes that were common to both inputs. For all data that passed the normality test, an un-paired 2-tailed T-test was used to test statistical significance. For data that did not pass the normality test, a non-parametric Mann-Whitney test was used. The statistical test used is indicated with the mean and SEM throughout the Results.

### Current-source density analysis

To reveal more precisely the spatial pattern of the major sites of synaptic activity, we calculated the current-source density (CSD) profile of visual and mechanosensory inputs along the neuropil as described in *Hamodi and Pratt (2015)* and *Mitzdorf and Singer (1977)* using a spatial differentiation grid of 20 μm. CSDs reveal current sinks, generated by positive current moving into the neuron, and sources, which are generated by positive current moving out of the neuron or negative current moving in through, for example, chloride channels. To generate CSDs, the second spatial derivative of FP traces was calculated by subtracting the trace obtained at a given position [V(x)] from the sum of the two flanking traces (i.e [V(x+n*Δx)]+ [V(x-n*Δx)]) and dividing by the square of the distance between the sites of recording (Δx). All calculations were performed using IGOR Pro and R software. Image plots were generated using Plotly software.

$$\frac{\partial^2 V}{\partial x^2} = \frac{V(x+n.\Delta x) + V(x-n.\Delta x) - 2V(x)}{(n.\Delta x^2)}$$

### Axon labelling

To image axons, DiI (1,1′ -dioctadecyl- 3,3,3′,3′ -tetramethylindocarbocyanine perchlorate; Molecular Probes) and DiD (1,1′-dioctadecyl-3,3,3′,3′-tetra- thylindodicarbocyanine perchlorate; Molecular Probes) were dissolved into 5% ethanol for injection. Xenopus tadpoles were anesthetized in 0.01% MS-222, DiI was injected into both sides of the hindbrain, and DiD was injected into both eyes, then tadpoles were fixed in 4% paraformaldehyde (PFA) for six days. Prior to imaging, samples were rinsed three times in 0.1 M phosphate buffer (PB: pH 7.4), mounted onto a customized slide for in vivo imaging, and then imaged using a laser scanning confocal microscope (Carl Zeiss LSM 710).

### Quantification of axon labeling

ImageJ software was used to quantify the pixel intensity of RGC and HB axon terminations in the laminar axis. For this, a single line was drawn parallel to the laminar axis at a point along the rostro-caudal axis where the neuropil is the widest. The fluorescent profile along this line was generated. We identified the distance from the distal edge of the laminar axis at which the peak pixel intensity, corresponding to the bulk of axonal input, of RGC and HB axons occurred and measured the distance (i.e. the gap) between the two peaks. The merged RGC and HB axon fluorescent profiles were also used to measure the amount of overlap between the two sets of axons by quantifying the area of each profile that was common to both inputs.

## Acknowledgements

We thank Michael Dillon for the R script for calculating second spatial derivatives. We also thank Carlos Aizenman for helpful comments on the manuscript.

## Additional information

### Funding

| Funder | Grant reference number | Author |
|---|---|---|
| Office of Experimental Program to Stimulate Competitive Research | (Outside the Box) Grant number 4201-11951-1001498 G | Zhenyu Liu<br>Kara G Pratt |
| National Institute of General Medical Sciences | P30-GM-32128 | Ali S Hamodi<br>Zhenyu Liu<br>Kara G Pratt |

The funders had no role in study design, data collection and interpretation, or the decision to submit the work for publication.

### Author contributions

ASH, KGP, Conception and design, Acquisition of data, Analysis and interpretation of data, Drafting or revising the article, Contributed unpublished essential data or reagents; ZL, Acquisition of data, Analysis and interpretation of data, Drafting or revising the article, Contributed unpublished essential data or reagents

### Author ORCIDs

Kara G Pratt, http://orcid.org/0000-0002-6743-4757

### Ethics

Animal experimentation: All experimental protocols have been approved by the University of Wyoming's Institutional Animal Care and Use Committee (IACUC). The protocol (# 20140411KP00089-03) was approved 04/11/16 to 04/10/17.

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
