## [Decision Letter]

Thank you for submitting your article "An NMDA receptor-dependent mechanism for subcellular segregation of sensory inputs in the tadpole optic tectum" for consideration by *eLife*. Your article has been reviewed by three peer reviewers, one of whom, Ronald Calabrese, is a member of our Board of Reviewing Editors and the evaluation has been overseen by Eve Marder as the Senior Editor.

The reviewers have discussed the reviews with one another and the Reviewing Editor has drafted this decision to help you prepare a revised submission.

Summary:

This study combines neuronal labeling and electrophysiology (LFP and CSD analysis) to study the segregation of visual and somatosensory inputs in the tectum of *Xenopus* tadpoles. The study confirms that these inputs are segregated onto the more distal and more proximal portions respectively of the same dendrites of tectal neurons. It is particularly clearly shown that they form functional synapses on these specific portions of the dendrites. The authors then show that elimination of the visual input causes spread of the somatosensory inputs and that normal segregation is compromised if the tadpoles are reared from the stage of initial synapse formation in the presence of NMDR blocker (MK-801) but not if the blockade is initiated at a later stage. Because the inputs are segregated initially during development, the results suggest a progression of activity-independent, then -dependent mechanisms that together result in the segregation of inputs across tectal neuron dendrites. Activity – specifically NMDAR-dependent activity – segregates the different inputs, implicating a correlation-based mechanism. This work thus adds to our knowledge about how inputs are segregated onto dendrites during development.

Essential revisions:

There are some concerns that must be addressed, but overall this appears to be a strong study of general significance.

1) The examples of anatomical data presented seem to support the conclusions, but the results must be quantified and comparisons made across a number of animals to test whether the differences seen in the examples presented do indeed stand up. Such analysis is necessary for the data in Figure 1. This concern is fleshed out more in the attached comments of reviewer #2.

2) The analysis of CSD sinks should be more thorough or this measure dropped and the physiological analysis restricted LFP. This concern is fleshed out more in the attached comments of reviewer #2.

3) A weakness of the paper is that the MK801 effect is the only direct evidence implicating NMDARs. Without evidence showing that disrupting NMDAR function specifically in tectal cells is sufficient to replicate the MK801 results the exact site of action is not clear. There are several reports of presynaptic NMDARs playing important roles, so the experiments provided do not exclude other NMDAR-dependent mechanisms being at play. Furthermore, in the discussion much attention is given to the fact that the data suggest a fundamentally different mechanism for the sorting of visual/non-visual input in *Xenopus* compared to other species. This makes the findings interesting, but also means that the cellular mechanism is an important one to hash out.

At the minimum this should be addressed in the discussion as a limitation of the preparation and something that should be addressed in the future. This would also be a good opportunity to provide any evidence (references or unpublished data) supporting a model in which NMDAR activity in tectal neurons is, in fact, the cellular mechanism mediating this phenomenon.

Reviewer #2:

While the interpretation is interesting, the data analysis is incomplete, decreasing confidence in the conclusions.

There are two principal concerns with the data analysis:

1) The anatomical pattern of innervation from the retina and the HB to the tectum, and the effect of eye enucleation and pharmacological treatment, are described anecdotally (a couple of examples). It is possible and probably essential to quantify (and summarize) the effect of enucleation and MK-801 on tectal innervation using some kind of laminar area measure or segregation measure (as is typically done for retino-geniculate projections or retino-collicular projections from either eye). This would put on much more solid ground the apparent effect of enucleation or MK-801 on HB projection 'segregation'.

2) The presentation of CSD and FP is a little confusing. Both are presented in the figures. It is argued that the CSD shows the site of sources and sinks (concentrated synaptic contacts occur at sinks if the structure is laminar in nature). However, only the FP amplitude is quantified, not the amplitude of the CSD sinks. In my opinion, the authors should also quantify the location (and spread as a result of enucleation) of the CSD sinks. If this is altered by the enucleation and/or MK-801, it would strengthen their arguments about the NMDA-receptor dependent 'competitive' segregation process. If quantification of the CSD pattern (and it's changes) is not possible, then this should be explained (and probably the CSD data should be de-emphasized or removed).

[Editors' note: further revisions were requested prior to acceptance, as described below.]

Thank you for submitting your article "An NMDA receptor-dependent mechanism for subcellular segregation of sensory inputs in the tadpole optic tectum" for consideration by *eLife*. Your article has been reviewed by one peer reviewer, and the evaluation has been overseen by a Reviewing Editor and Eve Marder as the Senior Editor. The reviewer has opted to remain anonymous.

The reviewer has discussed the reviews with the Reviewing Editor and the Reviewing Editor has drafted this decision to help you prepare a revised submission.

Summary:

The revision is very complete with most points forthrightly addressed. One more point remains. This revision should be quick and will not necessitate re-review. See comments of Reviewer #2

Reviewer #2:

The authors nicely included a quantification of the anatomical patterns of projection from the retina and HB to the tectum. This was done primarily by identifying the location of the peak of flourescence of the HB or retinal inputs to the tectum. While this measure does show a shift that is consistent with the interpretation of the authors, it would have been better to also include a measure of OVERLAP of the HB and retinal input to the tectum. That is, to what degree do the 'red' and 'green' signals from the retinal and HB axons overlap in the tectum with and without manipulation (enucleation, pharmacology). Such an OVERLAP index (instead of a measure of the location of the peak flourescence) more closely conforms to the interpretation of the anatomical and functional data the authors are presenting. I don't think this is an essential omission, but including an OVERLAP index would have been better. If an OVERLAP index does not show an effect consistent with their interpretation, perhaps the authors can comment on this in the discussion.

---

## [Author Response]

*Essential revisions:*

*There are some concerns that must be addressed, but overall this appears to be a strong study of general significance.*

*1) The examples of anatomical data presented seem to support the conclusions, but the results must be quantified and comparisons made across a number of animals to test whether the differences seen in the examples presented do indeed stand up. Such analysis is necessary for the data in Figure 1,G. This concern is fleshed out more in the attached comments of reviewer #2.*

See reviewer 2 #1 below.

*2) The analysis of CSD sinks should be more thorough or this measure dropped and the physiological analysis restricted LFP. This concern is fleshed out more in the attached comments of reviewer #2.*

See reviewer 2 #2 below.

3) A weakness of the paper is that the MK801 effect is the only direct evidence implicating NMDARs. Without evidence showing that disrupting NMDAR function specifically in tectal cells is sufficient to replicate the MK801 results the exact site of action is not clear. There are several reports of presynaptic NMDARs playing important roles, so the experiments provided do not exclude other NMDAR-dependent mechanisms being at play. Furthermore, in the discussion much attention is given to the fact that the data suggest a fundamentally different mechanism for the sorting of visual/non-visual input in Xenopus compared to other species. This makes the findings interesting, but also means that the cellular mechanism is an important one to hash out.

*At the minimum this should be addressed in the discussion as a limitation of the preparation and something that should be addressed in the future. This would also be a good opportunity to provide any evidence (references or unpublished data) supporting a model in which NMDAR activity in tectal neurons is, in fact, the cellular mechanism mediating this phenomenon.*

Whether segregation of visual and mechanosensory inputs relies specifically on postsynaptic NMDA receptors, presynaptic NMDA receptors, or both is not determined in this study, but will be addressed in the future. The following has been added to the Discussion:

“Exposing tadpoles to MK-801 blocks all NMDARs, so the possibility that presynaptic NMDARs could play a role in axon segregation cannot be ruled out. […] In addition, this will be an especially interesting experiment given that postsynaptic activation of CaMKII in cultured cortical neurons has been shown to remodel inputs such that the strongest inputs are kept and strengthened while other inputs are completely lost (Pratt et al., 2003).”

We do not suggest a different mechanism in tadpole compared to other species, but a different situation. In the tadpole tectum it is well established that single tectal neurons receive visual and mechanosensory input. Thus, our findings suggest a classic NMDAR-dependent correlation-driven mechanism underlies this subcellular segregation of inputs. This does not mean that NMDAR-dependent mechanisms are not involved in axon segregation in the optic tectum of Zebrafish or chick where axons are segregated, to the best of our knowledge, at the cellular (not subcellular) level. Indeed, NMDARs have been shown to be required for the proper segregation of RGC input onto different cells of the LGN, and for refinement of many circuits (refinement which involves axon targeting at the cellular, not subcellular, level).

*Reviewer #2:*

*While the interpretation is interesting, the data analysis is incomplete, decreasing confidence in the conclusions.*

*There are two principal concerns with the data analysis:*

*1) The anatomical pattern of innervation from the retina and the HB to the tectum, and the effect of eye enucleation and pharmacological treatment, are described anecdotally (a couple of examples). It is possible and probably essential to quantify (and summarize) the effect of enucleation and MK-801 on tectal innervation using some kind of laminar area measure or segregation measure (as is typically done for retino-geniculate projections or retino-collicular projections from either eye). This would put on much more solid ground the apparent effect of enucleation or MK-801 on HB projection 'segregation'.*

The manuscript now includes analysis of the spatial fluorescence intensity profiles of the HB and RGC axon projections across the tectum. ImageJ was used to generate the fluorescence intensity profiles (8-bit gray values). For the enucleation experiment, the peak fluorescence of HB inputs – which presumably reflects the peak in axon density – was measured and reported as the distance from the distal edge of the tectum. For the MK801 experiments, the distance from distal edge of peak fluorescence for both HB and RGC axons was measured as well as the distance (gap) separating these peaks. These data have been added to the figures and the text. The anatomical data are (strikingly) consistent with the FP and CSD data.

*2) The presentation of CSD and FP is a little confusing. Both are presented in the figures. It is argued that the CSD shows the site of sources and sinks (concentrated synaptic contacts occur at sinks if the structure is laminar in nature). However, only the FP amplitude is quantified, not the amplitude of the CSD sinks. In my opinion, the authors should also quantify the location (and spread as a result of enucleation) of the CSD sinks. If this is altered by the enucleation and/or MK-801, it would strengthen their arguments about the NMDA-receptor dependent 'competitive' segregation process. If quantification of the CSD pattern (and it's changes) is not possible, then this should be explained (and probably the CSD data should be de-emphasized or removed).*

The location of the major CSD sink along the laminar axis has been quantified and this data has been added to the figures and text. The CSD data is in accordance with the FP data. The amount of shift observed in the major CSD sink due to NMDAR blockade is the same amount of shift observed with the FPs. Identifying the major CSD sink was straightforward. We did not include the quantification of the amount of spread of the CSD sinks, however, as we did not observe such a pattern, probably because the CSDs reflect only δ. So for example if there was a big string of depolarization across the axis (i.e. a spread of depolarization) – this would not be reflected in the CSDs since a persistent depolarization (across space) would not be detected as a change in potential.

[Editors' note: further revisions were requested prior to acceptance, as described below.]

*Reviewer #2:*

*The authors nicely included a quantification of the anatomical patterns of projection from the retina and HB to the tectum. This was done primarily by identifying the location of the peak of flourescence of the HB or retinal inputs to the tectum. While this measure does show a shift that is consistent with the interpretation of the authors, it would have been better to also include a measure of OVERLAP of the HB and retinal input to the tectum. That is, to what degree do the 'red' and 'green' signals from the retinal and HB axons overlap in the tectum with and without manipulation (enucleation, pharmacology). Such an OVERLAP index (instead of a measure of the location of the peak flourescence) more closely conforms to the interpretation of the anatomical and functional data the authors are presenting. I don't think this is an essential omission, but including an OVERLAP index would have been better. If an OVERLAP index does not show an effect consistent with their interpretation, perhaps the authors can comment on this in the discussion.*

The amount of OVERLAP between the RGC and HB axons was calculated by measuring the area of the spatial fluorescent profile that was common to both sets of axons. These data are summarized in Figure 8, and the results are integrated into the Results section. The amount of axon overlap increased, compared to control, in both early and late MK801 manipulations. So, even though the late MK801 manipulation does not appear to alter the spatial position of peak field potential amplitude, peak CSD sink, nor peak axon fluorescence, it is somehow causing a subset of axons to desegregate which is resulting in increased overlap. We propose this reflects a relatively immature population of axons which are still relying on NMDA receptor-dependent mechanisms in the stage 45 tectum. This possibility is discussed in a few added sentences in the Discussion section.